# Turbulent enhancement of radar reflectivity factor for polydisperse cloud droplets

Keigo Matsuda[1] and Ryo Onishi[1]

[1]Center for Earth Information Science and Technology (CEIST), Japan Agency for Marine-Earth Science and Technology (JAMSTEC), 3173-25 Showa-machi, Kanazawa-ku, Yokohama 236-0001, Japan

*Correspondence to:* Keigo Matsuda (k.matsuda@jamstec.go.jp)

**Abstract.** The radar reflectivity factor is important for estimating cloud microphysical properties; thus, in this study, we determine the quantitative influence of microscale turbulent clustering of polydisperse droplets on the radar reflectivity factor. The theoretical solution for particulate Bragg scattering is obtained without assuming monodisperse droplet sizes. The scattering intensity is given by an integral function including the cross spectrum of number density fluctuations for two different droplet sizes. We calculate the cross spectrum based on turbulent clustering data, which are obtained by the direct numerical simulation (DNS) of particle-laden homogeneous isotropic turbulence. The results show that the coherence of the cross spectrum is close to unity for small wavenumbers and decreases almost exponentially with increasing wavenumber. This decreasing trend is dependent on the combination of Stokes numbers. A critical wavenumber is introduced to characterize the exponential decrease of the coherence and parametrized using the Stokes number difference. Comparison with DNS results confirms that the proposed model can reproduce the $r_p^3$-weighted power spectrum, which is proportional to the clustering influence on the radar reflectivity factor, to a sufficiently high accuracy. Furthermore, the proposed model is extended to incorporate the gravitational settling influence by modifying the critical wavenumber based on the analytical equation derived for the bidisperse radial distribution function. The estimate of the modified model also show good agreement with the DNS results for the case with gravitational droplet settling. The model is then applied to high-resolution cloud-simulation data obtained from a spectral-bin cloud simulation. The result shows that the influence of turbulent clustering can be significant inside turbulent clouds. The large influence is observed at the near-top of the clouds, where the liquid water content and the energy dissipation rate are sufficiently large.

## 1 Introduction

Radar remote sensing is widely used for observing a spatial distribution of cloud and precipitation particles because it can also provide estimates of cloud microphysical properties. The remote sensing data are analyzed and displayed using the radar reflectivity factor (mm$^6$ m$^{-3}$), $Z$, which is obtained by the following radar equation:

$$P_r = \frac{P_t G A_e k_m^4 |K|^2 V}{4^4 R^4} Z,$$ (1)

where $P_r$ and $P_t$ are the received and transmitted microwave powers, respectively, $G$ is the antenna gain, $A_e$ is the effective aperture of the antenna, $k_m$ is the microwave wavenumber, $K$ is the dielectric coefficient of a water droplet, $V$ is the measurement volume, and $R$ is the distance between the antenna and the cloud. The relationship between the radar reflectivity factor

and cloud microphysical properties is usually expressed based on the assumption of incoherent scattering (Gossard and Strauch, 1983). Incoherent scattering implies random and uniform dispersion of cloud droplets (Bohren and Huffman, 1983). For this case, the factor is proportional to the sum of the scattering intensities of the individual droplets in the measurement volume. In contrast, droplets form a nonuniform spatial distribution in turbulence, i.e. inertial particles concentrate in small-enstrophy

regions during turbulence due to the centrifugal motion (Maxey, 1987; Squires and Eaton, 1991; Wang and Maxey, 1993; Chen et al., 2006). This preferential concentration is often referred to as turbulent clustering. Nonuniform distribution of cloud droplets results in coherent scattering, which is also referred to as particulate Bragg scattering (Kostinski and Jameson, 2000). For this case, the interference of microwaves scattered by nonuniformly distributed droplets increases the scattered microwave intensity; i.e., the radar reflectivity factor increases due to particulate Bragg scattering. It should be noted that Bragg scattering

often indicates coherent scattering due to a nonuniform distribution of the refractive index of clear air. In the troposphere, turbulent mixing of temperature and water vapor cause spatial variation of the refractive index. To distinguish this effect from the particulate Bragg scattering, it is specifically referred to as clear-air Bragg scattering. The radar reflectivity factor for both Bragg scattering is dependent on the microwave frequency $f_{\mathrm{m}}$ ($= k_{\mathrm{m}}c/2\pi$, where $c$ is the speed of light), while the factor for incoherent scattering is independent of $f_{\mathrm{m}}$. Knight and Miller (1998) reported radar frequency dependence of their observation

results for small warm cumulus clouds using S- and X-band microwaves, which have wavelengths of 10 and 3 cm, respectively. Their observation data of S-band radar show a characteristic echo pattern of the mantle echo, in which strong radar echo was observed in cloud edges while it is relatively weak in cloud core regions. The mantle echo can be explained by clear-air Bragg scattering since the radar reflectivity factor difference is about 19 dB as it is expected for ideal Bragg scattering. They also reported that there are many cases where frequency dependence is not explained by clear-air Bragg scattering. In such cases,

the S-band radar echo is about 10 dB stronger than the X-band radar echo, and the difference is observed also in the cloud core regions. Rogers and Brown (1997) also reported similar frequency dependence of radar returns during their observation of a smoke plume from intense fire, using a UHF wind profiler and an X-band radar. Erkelens et al. (2001) analyzed the observation data and estimated the influence of coherent scattering by cloud droplets using the $-5/3$ power law of turbulent spectrum, which represents turbulent mixing of cloud water with environmental clear air; i.e., turbulent entrainment. They concluded that

coherent scattering by cloud droplets can be more significant than clear-air Bragg scattering, whereas turbulent entrainment is not the only factor relevant to the frequency dependence in the observation data. Kostinski and Jameson (2000) first pointed out the possibility that particulate Bragg scattering due to turbulent clustering leads to the frequency dependence reported by Knight and Miller (1998). To evaluate the quantitative influence of turbulent clustering on the radar reflectivity factor, it is crucial to understand the spatial structure of turbulent clustering. Turbulent clustering has been discussed in many literatures

because it can enhance the collision growth of cloud droplets (e.g., Sundaram and Collins, 1997; Reade and Collins, 2000; Ayala et al., 2008a, b; Onishi et al., 2009; Wang et al., 2009; Onishi and Vassilicos, 2014), and statistical data on turbulent clustering have been obtained for scales relevant to droplet collisions. However, these data cannot be adopted for particulate Bragg scattering because the clustering scales relevant to particulate Bragg scattering are on the microwave wavelength, which is larger than droplet collision scales. Quantitative estimate of particulate Bragg scattering due to turbulent clustering is first

provided by Dombrovsky and Zaichik (2010). Their analytical estimate was based on a clustering model for droplet collision

scales but indicated that turbulent clustering can lead to considerable increase in the radar reflectivity factor. Matsuda et al. (2014) clarified the quantitative influence based on turbulent clustering data obtained by a three-dimensional direct numerical simulation (DNS), which covered the clustering scales on the microwave wavelength. They estimated the increase in the radar reflectivity factor due to turbulent clustering by calculating the power spectrum of number density fluctuations $E_{np}(k|r_p)$, where $k$ is the wavenumber and $r_p$ is the droplet radius. The power spectrum $E_{np}(k|r_p)$ is strongly dependent on the droplet size: More specifically, $E_{np}(k|r_p)$ is dependent on the Stokes number, St, which is defined as $\mathrm{St} \equiv \tau_p/\tau_\eta$ ($\tau_p$ is the relaxation time of droplet motion and $\tau_\eta$ is the Kolmogorov time). However, the discussion of radar reflectivity factor increases is limited to cases of monodisperse particles. Thus, the results are not directly applicable to particulate Bragg scattering for real cloud systems, in which cloud droplets have broad droplet size distributions.

Therefore, this study aims to investigate the influence of turbulent clustering of polydisperse droplets on particulate Bragg scattering. Firstly, the theoretical formulation of particulate Bragg scattering is extended for polydisperse particles and expressed using the cross spectrum of number density fluctuations for two different droplet sizes. Secondly, the three-dimensional DNS of particle-laden homogeneous isotropic turbulence is performed to obtain turbulent droplet clustering data, which are used to calculate the power spectrum and the cross spectrum of number density fluctuations. A parameterization for the cross spectrum is then proposed considering the dependence of the Stokes number combination, and the influence of gravitational settling is discussed and incorporated. Finally, in order to investigate the impact of turbulent clustering on radar observations of realistic clouds, the proposed model is applied to high-resolution cloud-simulation data obtained by a spectral-bin cloud microphysics simulation.

## 2 Theory

Here, we aim to formulate the radar reflectivity factor $Z$ for a nonuniform distribution of polydisperse cloud droplets based on the discussion of Gossard and Strauch (1983), but without assuming monodisperse droplet sizes. Because the radii of cloud droplets are much smaller than the microwave wavelength, the electric field vector of the microwaves scattered by a single droplet, $\mathbf{E}_{sca}(t, \mathbf{x}, r_p)$, is given by the Rayleigh scattering approximation:

$$\mathbf{E}_{sca}(t, \mathbf{x}, r_p) = \mathbf{E}_{inc} \frac{k_m^2 K r_p^3}{R} \sin\chi \exp\left[i\left\{\omega t - \mathbf{k}_{sca} \cdot (\mathbf{x}_r - \mathbf{x}) - \mathbf{k}_{inc} \cdot (\mathbf{x} - \mathbf{x}_t)\right\}\right], \tag{2}$$

where $\mathbf{E}_{inc}$ is the electric field amplitude vector of the incident microwave, $r_p$ is the droplet radius, $\mathbf{x}$, $\mathbf{x}_t$, and $\mathbf{x}_r$ are the position vectors of the droplet, microwave transmitter, and microwave receiver, respectively, $\mathbf{k}_{inc}$ and $\mathbf{k}_{sca}$ are the wavenumber vectors of the incident and scattered microwaves, respectively, which satisfy $|\mathbf{k}_{inc}| = |\mathbf{k}_{sca}| = k_m$, and $\chi$ is the angle between $\mathbf{E}_{inc}$ and $\mathbf{k}_{sca}$. Considering the droplet-size dependence of $\mathbf{E}_{sca}(t, \mathbf{x}, r_p)$, the electric power of the microwave scattered by a group of droplets, $P_s$, is given by

$$P_s = \left| \overline{\int_{\mathbf{x} \in V} \int_0^\infty \mathbf{E}_{sca}(t, \mathbf{x}, r_p) n(\mathbf{x}, r_p) dr_p d\mathbf{x}} \right|^2 / \zeta, \tag{3}$$

where $n(\mathbf{x}, r_\mathrm{p}) dr_\mathrm{p} d\mathbf{x}$ is the number of droplets with radii from $r_\mathrm{p}$ to $r_\mathrm{p} + dr_\mathrm{p}$ in an infinitesimal volume $d\mathbf{x}$ at position $\mathbf{x}$, and $\zeta$ is the intrinsic impedance. The overbar denotes the temporal average. The relationship between the radar reflectivity factor $Z$ and the scattering properties of target clouds is given by

$$Z = \frac{\lambda^4}{\pi^5 |K|^2 V \sin^2 \chi} \sigma, \tag{4}$$

where $\lambda$ is the microwave wavelength and $\sigma$ is the radar cross section (Gossard and Strauch, 1983), which is defined as

$$\sigma = 4\pi R^2 \frac{P_\mathrm{s}}{P_\mathrm{o}}, \tag{5}$$

where $P_\mathrm{o}$ is the electric power of the incident microwave, which is given by $P_\mathrm{o} = |\mathbf{E}_\mathrm{inc}|^2 / \zeta$. Substitution of Eqs. (2), (3), and (5) into Eq. (4) yields

$$Z = \frac{2^6}{V} \overline{\left| \int_{\mathbf{x} \in V} \int_0^\infty r_\mathrm{p}^3 n(\mathbf{x}, r_\mathrm{p}) \exp\left(-i\boldsymbol{\kappa} \cdot \mathbf{x}\right) dr_\mathrm{p} d\mathbf{x} \right|^2}, \tag{6}$$

where the wavenumber vector $\boldsymbol{\kappa}$ is defined as $\boldsymbol{\kappa} = \mathbf{k}_\mathrm{inc} - \mathbf{k}_\mathrm{sca}$. Note that radar remote sensing typically uses backward scattering; i.e., $\mathbf{k}_\mathrm{sca} = -\mathbf{k}_\mathrm{inc}$; thus, $\boldsymbol{\kappa} = 2\mathbf{k}_\mathrm{inc}$.

Similarly to Gossard and Strauch (1983), we assume $n(\mathbf{x}, r_\mathrm{p})$ to be composed of the temporal-average and fluctuation parts; i.e., $n(\mathbf{x}, r_\mathrm{p}) = \overline{n(\mathbf{x}, r_\mathrm{p})} + \delta n(\mathbf{x}, r_\mathrm{p})$. The temporal-average part, $\overline{n(\mathbf{x}, r_\mathrm{p})}$, contributes to the separated reflection; therefore, the contribution of this part is negligibly small when $\overline{n(\mathbf{x}, r_\mathrm{p})}$ has no fluctuation at a spatial scale of half the wavelength (Erkelens

et al., 2001). Thus, we neglect the contribution of $\overline{n(\mathbf{x}, r_\mathrm{p})}$. Then, we obtain

$$Z = 2^6 \int_0^\infty \int_0^\infty \left\{ r_\mathrm{p}^3 r_\mathrm{p}'^3 \int_\mathbf{r} \langle \delta n(\mathbf{x}, r_\mathrm{p}) \delta n(\mathbf{x} + \mathbf{r}, r_\mathrm{p}') \rangle \exp\left(-i\boldsymbol{\kappa} \cdot \mathbf{r}\right) d\mathbf{r} \right\} dr_\mathrm{p} dr_\mathrm{p}', \tag{7}$$

where the angular brackets represent a temporal and spatial average in the measurement volume.

In order to decompose the spatial correlation function $\langle \delta n(\mathbf{x}, r_\mathrm{p}) \delta n(\mathbf{x} + \mathbf{r}, r_\mathrm{p}') \rangle$, we introduce the probability density function (PDF) of droplet radius $r_\mathrm{p}$ to the measurement volume, $q_\mathrm{r}(r_\mathrm{p})$, and the number density distribution function for monodisperse

droplets with a radius of $r_\mathrm{p}$, $n_\mathrm{p}(\mathbf{x}|r_\mathrm{p})$. The PDF is defined as $q_\mathrm{r}(r_\mathrm{p}) \equiv \frac{1}{N_\mathrm{p}} \int_{\mathbf{x} \in V} \overline{n(\mathbf{x}, r_\mathrm{p})} d\mathbf{x}$, where $N_\mathrm{p}$ is the total number of droplets in the measurement volume; i.e., $N_\mathrm{p} \equiv \int_0^\infty \int_{\mathbf{x} \in V} \overline{n(\mathbf{x}, r_\mathrm{p})} d\mathbf{x} dr_\mathrm{p}$. The PDF satisfies $\int_0^\infty q_\mathrm{r}(r_\mathrm{p}) dr_\mathrm{p} = 1$. The number density distribution function for monodisperse droplets is then defined as $n_\mathrm{p}(\mathbf{x}|r_\mathrm{p}) \equiv n(\mathbf{x}, r_\mathrm{p})/q_\mathrm{r}(r_\mathrm{p})$ so that $n(\mathbf{x}, r_\mathrm{p})$ is given by $n(\mathbf{x}, r_\mathrm{p}) = n_\mathrm{p}(\mathbf{x}|r_\mathrm{p}) q_\mathrm{r}(r_\mathrm{p})$. The number density distribution function $n_\mathrm{p}(\mathbf{x}|r_\mathrm{p})$ satisfies $\int_{\mathbf{x} \in V} \overline{n_\mathrm{p}(\mathbf{x}|r_\mathrm{p})} d\mathbf{x} = N_\mathrm{p}$ for arbitrary $r_\mathrm{p}$. Note that the spatial correlation function $\langle \delta n(\mathbf{x}, r_\mathrm{p}) \delta n(\mathbf{x} + \mathbf{r}, r_\mathrm{p}') \rangle$ for $r_\mathrm{p}' = r_\mathrm{p}$ is discontinuous at $\mathbf{r} = \mathbf{0}$ because the

droplet distribution is composed of spatially discrete points. The singularity is given by $\langle n(\mathbf{x}, r_\mathrm{p}) \rangle \delta(\mathbf{r}) \delta(r_\mathrm{p}' - r_\mathrm{p})$, where $\delta(\mathbf{r})$ and $\delta(r_\mathrm{p}' - r_\mathrm{p})$ are the Dirac delta functions. Thus, the spatial correlation function is given by

$$\langle \delta n(\mathbf{x}, r_\mathrm{p}) \delta n(\mathbf{x} + \mathbf{r}, r_\mathrm{p}') \rangle = \langle n_\mathrm{p} \rangle \delta(\mathbf{r}) q_\mathrm{r}(r_\mathrm{p}) \delta(r_\mathrm{p}' - r_\mathrm{p}) + \Psi(\mathbf{r}|r_\mathrm{p}, r_\mathrm{p}') q_\mathrm{r}(r_\mathrm{p}) q_\mathrm{r}(r_\mathrm{p}'), \tag{8}$$

where $\langle n_{\mathrm{p}} \rangle$ is the averaged number density ($\langle n_{\mathrm{p}} \rangle \equiv N_{\mathrm{p}}/V$) and $\Psi(\mathbf{r}|r_{\mathrm{p}}, r'_{\mathrm{p}})$ is defined as the continuous part of $\langle \delta n_{\mathrm{p}}(\mathbf{x}|r_{\mathrm{p}}) \delta n_{\mathrm{p}}(\mathbf{x}+ \mathbf{r}|r'_{\mathrm{p}}) \rangle$. Substitution of Eq. (8) into Eq. (7) and adoption of the isotropic clustering assumption (Gossard and Strauch, 1983) yield

$$Z = 2^6 \langle r_{\mathrm{p}}^6 \rangle \langle n_{\mathrm{p}} \rangle + 2^7 \pi^2 \kappa^{-2} E_{\mathrm{r3np}}(\kappa), \tag{9}$$

where $\kappa$ is $\kappa = |\boldsymbol{\kappa}|$, $\langle r_{\mathrm{p}}^6 \rangle$ is given by $\langle r_{\mathrm{p}}^6 \rangle = \int_0^\infty r_{\mathrm{p}}^6 q_{\mathrm{r}}(r_{\mathrm{p}}) dr_{\mathrm{p}}$, and $E_{\mathrm{r3np}}(k)$ is the $r_{\mathrm{p}}^3$-weighted power spectrum, defined as

$$E_{\mathrm{r3np}}(k) \equiv \int\limits_0^\infty \int\limits_0^\infty r_{\mathrm{p}}^3 r_{\mathrm{p}}'^3 q_{\mathrm{r}}(r_{\mathrm{p}}) q_{\mathrm{r}}(r'_{\mathrm{p}}) C_{\mathrm{np}}(k|r_{\mathrm{p}}, r'_{\mathrm{p}}) dr_{\mathrm{p}} dr'_{\mathrm{p}}, \tag{10}$$

where $C_{\mathrm{np}}(k|r_{\mathrm{p}}, r'_{\mathrm{p}})$ is the cross spectrum of number density fluctuations for $n_{\mathrm{p}}(\mathbf{x}|r_{\mathrm{p}})$ and $n_{\mathrm{p}}(\mathbf{x}|r'_{\mathrm{p}})$: The cross spectrum $C_{\mathrm{np}}(k|r_{\mathrm{p}}, r'_{\mathrm{p}})$ is defined as $C_{\mathrm{np}}(k|r_{\mathrm{p}}, r'_{\mathrm{p}}) \equiv \int_{|\mathbf{k}|=k} \widetilde{\Psi}(\mathbf{k}|r_{\mathrm{p}}, r'_{\mathrm{p}}) d\sigma_k$; i.e., the integration of $\widetilde{\Psi}(\mathbf{k}|r_{\mathrm{p}}, r'_{\mathrm{p}})$ over the spherical shell, $\sigma_k$, at $|\mathbf{k}| = k$, in which $\widetilde{\Psi}(\mathbf{k}|r_{\mathrm{p}}, r'_{\mathrm{p}})$ is the cross spectral density function, defined as

$$\widetilde{\Psi}(\mathbf{k}|r_{\mathrm{p}}, r'_{\mathrm{p}}) = \frac{1}{(2\pi)^3} \int\limits_{\mathbf{r}} \Psi(\mathbf{r}|r_{\mathrm{p}}, r'_{\mathrm{p}}) \exp(-i\mathbf{k} \cdot \mathbf{r}) d\mathbf{r}. \tag{11}$$

The first and second terms on the right hand side of Eq. (9) are the incoherent and coherent scattering parts, respectively; particulate Bragg scattering is caused by the second term. Eqs. (9) and (10) imply that the particulate Bragg scattering part of $Z$ for an arbitrary droplet size distribution can be calculated using the cross spectrum for bidisperse droplet size conditions. When droplets are distributed randomly and uniformly, the second term equals zero. Thus, the radar reflectivity factor when assuming a random and uniform droplet distribution is given by the first term; i.e., $Z_{\mathrm{incoh}} = 2^6 \langle r_{\mathrm{p}}^6 \rangle \langle n_{\mathrm{p}} \rangle$.

It should be noted that Eq. (9) satisfies the theoretical solution for particulate Bragg scattering of monodisperse droplets: For the case of monodisperse droplets with radii of $r_{\mathrm{p1}}$, the PDF of droplet radius is given by $q_{\mathrm{r}}(r_{\mathrm{p}}) = \delta(r_{\mathrm{p}} - r_{\mathrm{p1}})$. Then, the radar reflectivity factor $Z$ is given by

$$Z = 2^6 r_{\mathrm{p1}}^6 \langle n_{\mathrm{p}} \rangle + 2^7 \pi^2 \kappa^{-2} r_{\mathrm{p1}}^6 E_{\mathrm{np}}(\kappa|r_{\mathrm{p1}}), \tag{12}$$

where $E_{\mathrm{np}}(k|r_{\mathrm{p}})$ is the power spectrum of number density fluctuations, which satisfies $E_{\mathrm{np}}(k|r_{\mathrm{p}}) = C_{\mathrm{np}}(k|r_{\mathrm{p}}, r_{\mathrm{p}})$. Note that $E_{\mathrm{np}}(k|r_{\mathrm{p}})$ is defined as $E_{\mathrm{np}}(k|r_{\mathrm{p}}) \equiv \int_{|\mathbf{k}|=k} \widetilde{\Phi}(\mathbf{k}|r_{\mathrm{p}}) d\sigma_k$, where $\widetilde{\Phi}(\mathbf{k}|r_{\mathrm{p}})$ is the power spectral density function of $n_{\mathrm{p}}(\mathbf{x}|r_{\mathrm{p}})$, defined as

$$\widetilde{\Phi}(\mathbf{k}|r_{\mathrm{p}}) = \frac{1}{(2\pi)^3} \int\limits_{\mathbf{r}} \Psi(\mathbf{r}|r_{\mathrm{p}}, r_{\mathrm{p}}) \exp(-i\mathbf{k} \cdot \mathbf{r}) d\mathbf{r}. \tag{13}$$

## 3 Computational method

### 3.1 Direct numerical simulation

In order to obtain turbulent clustering data for calculating the cross spectrum, we have performed a three-dimensional DNS for particle-laden homogeneous isotropic turbulence. Three-dimensional incompressible turbulent air flows were calculated by

solving the continuity and Navier-Stokes equations:

$$\frac{\partial u_i}{\partial x_i} = 0, \tag{14}$$

$$\frac{\partial u_i}{\partial t} + \frac{\partial u_i u_j}{\partial x_j} = -\frac{1}{\rho_a}\frac{\partial p}{\partial x_i} + \nu \frac{\partial^2 u_i}{\partial x_j \partial x_j} + F_i, \tag{15}$$

where $u_i$ is the flow velocity in the $i$th direction, $p$ is the pressure, $\rho_a$ is the air density, $\nu$ is the kinematic viscosity, and $F_i$ is the external forcing term. The nonlinear term was discretized by the fourth-order central difference scheme (Morinishi et al., 1998). The time integration was calculated by the second-order Runge-Kutta scheme. The HSMAC method (Hirt and Cook, 1972) was used for velocity-pressure coupling. The external forcing was applied to maintain the intensity of large-scale eddies for wavenumbers $\mathbf{k}$ in the range $|\mathbf{k}L_0| < 2$ (Onishi et al., 2011), where $L_0$ is the representative length scale.

Droplet motions were simulated by Lagrangian point-particle tracking. Here, we assumed that the droplet density $\rho_p$ is much larger than $\rho_a$ and the drag term is given based on the Stokes law. The droplet motions were tracked by

$$\frac{dv_i}{dt} = -\frac{v_i - u_i}{\tau_p} + g_i, \tag{16}$$

where $v_i$ and $g_i$ are the particle velocity and gravitational acceleration in the $i$th direction, respectively. $\tau_p$ is the droplet relaxation time, which is given by

$$\tau_p = \frac{\rho_p}{\rho_a}\frac{2r_p^2}{9\nu}. \tag{17}$$

The effects of turbulent modulation and droplet collision were neglected for simplicity because these effects were typically small in the time scale of $\tau_\eta$ in clouds.

The computational domain was set as a cube with edge lengths of $2\pi L_0$. Periodic boundary conditions were applied in all three directions. A uniform staggered grid was used for discretization. The number of grid points was set to $512^3$. A Taylor microscale-based turbulent Reynolds number of the obtained flow was $\mathrm{Re}_\lambda = 204$, where $\mathrm{Re}_\lambda$ is defined as $\mathrm{Re}_\lambda \equiv l_\lambda u'/\nu$, where $l_\lambda$ is the Taylor microscale and $u'$ is the RMS value of the velocity fluctuation. Note that this value of $\mathrm{Re}_\lambda$ is sufficiently large to obtain turbulent clustering data for high Reynolds number turbulence in the wavenumber range relevant to radar observations (Matsuda et al., 2014) (see section 3.2). The kinematic viscosity, $\nu$, was set to $1.5 \times 10^{-5}$ m$^2$/s, and the ratio of the droplet density to the air density, $\rho_p/\rho_a$, was set to 840, assuming 1 atm and 298 K. The total number of droplets, $N_p$, was set to $1.5 \times 10^7$.

For this study, we performed the DNS for monodisperse and polydisperse droplets. Table 1 shows the computational settings for turbulence, droplet size, and gravitational acceleration. For monodisperse droplets, the droplet motions in an identical turbulent flow field were calculated for six values of Stokes number, $\mathrm{St}$. The clustering data for the monodisperse cases were used for calculating the cross spectrum of number density fluctuations for any combinations of these $\mathrm{St}$. For polydisperse droplets, a typical droplet size distribution for maritime cumulus clouds (the size distribution data named "CUMA" in Hess et al. (1998)) was applied, and the droplets were tracked in turbulent flows using three different energy dissipation rates $\epsilon$. $\epsilon$ values of the obtained turbulent flows were approximately 100, 400, and 1000 cm$^2$/s$^3$, which can be observed in cumulus and

**Table 1.** Computational settings of the DNS.

| Case | $L_0$ (m) | $\epsilon$ (cm$^2$/s$^3$) | Droplet size | $g$ (m/s$^2$) |
|---|---|---|---|---|
| St005_eps400 | 0.0682 | 395 | monodisperse (St = 0.05) | 0 |
| St01_eps400 | | | monodisperse (St = 0.10) | |
| St02_eps400 | | | monodisperse (St = 0.20) | |
| St05_eps400 | | | monodisperse (St = 0.50) | |
| St1_eps400 | | | monodisperse (St = 1.0) | |
| St2_eps400 | | | monodisperse (St = 2.0) | |
| CUMA_eps100 | 0.0961 | 100 | polydisperse (CUMA) | 0, 9.8 |
| CUMA_eps400 | 0.0682 | 395 | polydisperse (CUMA) | |
| CUMA_eps1000 | 0.0541 | 990 | polydisperse (CUMA) | |

cumulonimbus clouds (Pinsky et al., 2008). The data for the polydisperse droplet cases were used to discuss the reliability of the proposed cross spectrum model. It should be noted that the droplet size distribution for the polydisperse cases were identical but the Stokes number histograms were different; the Stokes numbers corresponding to the modal radius (10.4 $\mu$m) for $\epsilon = 100$, 400, and 1000 cm$^2$/s$^3$ were 0.035, 0.069, and 0.10, respectively. The gravitational acceleration $g \equiv \sqrt{g_i g_i}$ was set to zero for the monodisperse cases. The DNS for polydisperse droplets were performed under the conditions with and without gravitational settling. The Froude numbers, Fr ($\equiv a_\eta/g$, in which $a_\eta \equiv \epsilon^{3/4}\nu^{-1/4}$ is the Kolmogorov acceleration), for the cases with gravitational settling were 0.0520, 0.145, and 0.289 for $\epsilon = 100$, 400, and 1000 cm$^2$/s$^3$, respectively. The influence of gravitational settling on turbulent particle clustering is often discussed using the settling parameter, $S_v$, which is defined as $S_v \equiv v_T/u_\eta$ (e.g., Wang and Maxey, 1993; Grabowski and Vaillancourt, 1999; Bec et al., 2014; Ireland et al., 2016), where $v_T = \tau_p g$ is the terminal settling velocity and $u_\eta \equiv (\nu\epsilon)^{1/4}$ is the Kolmogorov velocity. Note that $S_v$ satisfies $S_v = St/Fr$. The settling parameters corresponding to the modal radius for $\epsilon = 100$, 400, and 1000 cm$^2$/s$^3$ were 0.67, 0.48, and 0.38, respectively.

### 3.2  Computation of power spectrum and cross spectrum

The power spectral density function $\widetilde{\Phi}(\mathbf{k}|r_p)$ and the cross spectral density function $\widetilde{\Psi}(\mathbf{k}|r_{p1}, r_{p2})$ are calculated from the Lagrangian droplet distribution data as follows:

$$\widetilde{\Phi}(\mathbf{k}|r_p) = L_0^{-3}\langle \widetilde{n_p}(\mathbf{k}|r_p)\widetilde{n_p}(-\mathbf{k}|r_p)\rangle, \tag{18}$$

$$\widetilde{\Psi}(\mathbf{k}|r_{p1}, r_{p2}) = L_0^{-3}\langle \widetilde{n_p}(\mathbf{k}|r_{p1})\widetilde{n_p}(-\mathbf{k}|r_{p2})\rangle, \tag{19}$$

where $\widetilde{n_p}(\mathbf{k}|r_p)$ is the Fourier component of the droplet number density distribution, $n_p(\mathbf{x}|r_p)$, and the angle brackets denote an ensemble average. The number density distribution for Lagrangian discrete droplets is given by

$$n_p(\mathbf{x}|r_p) = \sum_{j=1}^{N_p} \delta(\mathbf{x} - \mathbf{x}_{p,j}), \tag{20}$$

where $\mathbf{x}_{p,j}$ is the position vector of the $j$th droplet with radius $r_p$, and $N_p$ is the total number of droplets with radius $r_p$. The Fourier components of Eq. (20) are then given by

$$\widetilde{n_p}(\mathbf{k}|r_p) = \frac{1}{(2\pi)^3} \sum_{j=1}^{N_p} \exp(-i\mathbf{k} \cdot \mathbf{x}_{p,j}). \tag{21}$$

Substitution of Eq. (21) into Eq. (18) yields

$$\frac{\widetilde{\Phi}(\mathbf{k}|r_p)}{\langle n_p \rangle^2 L_0^3} = \frac{1}{N_p^2} \left\langle \sum_{j=1}^{N_p} \exp(-i\mathbf{k} \cdot \mathbf{x}_{p,j}) \sum_{j'=1, j' \neq j}^{N_p} \exp(i\mathbf{k} \cdot \mathbf{x}_{p,j'}) \right\rangle \tag{22}$$

$$= \frac{1}{N_p^2} \left[ \left\langle \left\{ \sum_{j=1}^{N_p} \cos(\mathbf{k} \cdot \mathbf{x}_{p,j}) \right\}^2 \right\rangle + \left\langle \left\{ \sum_{j=1}^{N_p} \sin(\mathbf{k} \cdot \mathbf{x}_{p,j}) \right\}^2 \right\rangle \right] - \frac{1}{N_p}. \tag{23}$$

Similarly, substitution of Eq. (21) into Eq. (19) yields

$$\frac{\widetilde{\Psi}(\mathbf{k}|r_{p1}, r_{p2})}{\langle n_{p1} \rangle \langle n_{p2} \rangle L_0^3} = \frac{1}{N_{p1} N_{p2}} \left\langle \sum_{j=1}^{N_{p1}} \exp(-i\mathbf{k} \cdot \mathbf{x}_{p1,j}) \sum_{j'=1}^{N_{p2}} \exp(i\mathbf{k} \cdot \mathbf{x}_{p2,j'}) \right\rangle \tag{24}$$

$$= \frac{1}{N_{p1} N_{p2}} \left\{ \left\langle \sum_{j=1}^{N_{p1}} \cos(\mathbf{k} \cdot \mathbf{x}_{p1,j}) \sum_{j'=1}^{N_{p2}} \cos(\mathbf{k} \cdot \mathbf{x}_{p2,j'}) \right\rangle + \left\langle \sum_{j=1}^{N_{p1}} \sin(\mathbf{k} \cdot \mathbf{x}_{p1,j}) \sum_{j'=1}^{N_{p2}} \sin(\mathbf{k} \cdot \mathbf{x}_{p2,j'}) \right\rangle \right\}, \tag{25}$$

where $\langle n_{p1} \rangle$ and $\langle n_{p2} \rangle$ are the average number density of droplets with radii of $r_{p1}$ and $r_{p2}$ (i.e., $\langle n_{p1} \rangle \equiv \langle n_p(\mathbf{x}|r_{p1}) \rangle$ and $\langle n_{p2} \rangle \equiv \langle n_p(\mathbf{x}|r_{p2}) \rangle$), respectively, $N_{p1}$ and $N_{p2}$ are the numbers of droplets with radii of $r_{p1}$ and $r_{p2}$, respectively, $\mathbf{x}_{p1,j}$ is the position of the $j$th droplet with a radius of $r_{p1}$, and $\mathbf{x}_{p2,j'}$ is the position of the $j'$th droplet with a radius of $r_{p2}$. It should be noted that the imaginary part of $\widetilde{\Psi}(\mathbf{k}|r_{p1}, r_{p2})$ is omitted in Eq. (25) because, statistically, it should be zero. We confirmed that the imaginary part of $C_{np}(k|r_{p1}, r_{p2})$ calculated from the DNS data is $O(10^{-4})$, which is caused by the statistical and truncation errors.

The spectral density functions, $\widetilde{\Phi}(\mathbf{k}|r_p)$ and $\widetilde{\Psi}(\mathbf{k}|r_{p1}, r_{p2})$, were calculated for discrete wavenumbers $\mathbf{k}L_0 = (h_1, h_2, h_3)$, where $h_1$, $h_2$, and $h_3$ are arbitrary integers, that satisfy $k - \Delta k/2 \leq |\mathbf{k}| < k + \Delta k/2$, where $\Delta k$ was set to $1/L_0$. $E_{np}(k|r_p)$ and $C_{np}(k|r_{p1}, r_{p2})$ were then obtained by the following equations:

$$E_{np}(k|r_p) = \frac{1}{\Delta k} \sum_{k - \frac{1}{2}\Delta k \leq |\mathbf{k}| < k + \frac{1}{2}\Delta k} \widetilde{\Phi}(\mathbf{k}|r_p), \tag{26}$$

$$C_{np}(k|r_{p1}, r_{p2}) = \frac{1}{\Delta k} \sum_{k - \frac{1}{2}\Delta k \leq |\mathbf{k}| < k + \frac{1}{2}\Delta k} \widetilde{\Psi}(\mathbf{k}|r_{p1}, r_{p2}). \tag{27}$$

The spectra, $E_{np}(k|r_p)$ and $C_{np}(k|r_{p1}, r_{p2})$, were calculated for 19 representative wavenumbers in the same way as Matsuda et al. (2014). The ensemble average in Eq. (23) was obtained by averaging 10 temporal slices of the droplet distributions, which were sampled for intervals of $T_0 = L_0/U_0$. The ensemble average in Eq. (25) was also obtained for 10 pairs of temporal slices. Each pair was composed of temporal slices at the same time step for different St cases.

Matsuda et al. (2014) normalized the power spectrum $E_{np}(k|r_p)$ by using $\langle n_p \rangle$ and the Kolmogorov scale, $l_\eta$, defined as $l_\eta \equiv \nu^{3/4}\epsilon^{-1/4}$, and confirmed that, for $\text{Re}_\lambda \geq 204$, the $\text{Re}_\lambda$ dependence of the normalized power spectrum is negligible at the wavenumber range relevant to radar observations ($0.05 < kl_\eta < 4.0$). Thus, this study used the same normalization for $E_{np}(k|r_p)$ and $C_{np}(k|r_{p1}, r_{p2})$ as follows:

$$E_{np}^*(\xi|\text{St}) = E_{np}^*(kl_\eta|\text{St}) \equiv \frac{E_{np}(k|r_p)}{\langle n_p \rangle^2 l_\eta}, \tag{28}$$

$$C_{np}^*(\xi|\text{St}_1, \text{St}_2) = C_{np}^*(kl_\eta|\text{St}_1, \text{St}_2) \equiv \frac{C_{np}(k|r_{p1}, r_{p2})}{\langle n_{p1} \rangle \langle n_{p2} \rangle l_\eta}, \tag{29}$$

where $\xi$ is the normalized wavenumber defined as $\xi \equiv kl_\eta$ and St, $\text{St}_1$, and $\text{St}_2$ are the Stokes numbers of water droplets with radii of $r_p$, $r_{p1}$, and $r_{p2}$, respectively.

## 4   DNS results and parameterization

### 4.1   Spatial droplet distribution and cross spectrum

Figure 1 shows the spatial distribution of droplets for $\text{St} = 0.2$, 0.5, and 1.0. Droplets located in the range of $0 < z < 4l_\eta$ are indicated. The droplet position data were sampled at the same time step; i.e., the background turbulent flow field is identical. Figure 1(a) is the overall view of the region with a size of $2\pi L_0 \times 2\pi L_0$, while Figure 1(b) is the magnified view for the region with a size of $0.5\pi L_0 \times 0.5\pi L_0$. The clusters and void areas are clearly observed for all St cases. Their location for $\text{St} = 0.5$ is almost the same as that for the other St. This is because the locations of clusters and void areas for small St are strongly

dependent on the instantaneous turbulent flow field. However, the small-scale structure of clusters is not exactly the same; i.e., the droplets become more concentrated in clusters as St increases. Figure 2 shows the power spectra $E_{np}^*(\xi|\text{St})$ and the cross spectra $C_{np}^*(\xi|\text{St}_1, \text{St}_2)$ obtained from the DNS data. Note that the high wavenumber portion is omitted when the value of the cross spectrum is smaller than the computational error level ($10^{-4}$). The power spectra $E_{np}^*(\xi|\text{St})$ show power law type slopes at wavenumbers smaller and larger than the peak location. The peak height and slope of the spectra are strongly dependent

on the Stokes number; this Stokes number dependence was discussed by Matsuda et al. (2014). In the small wavenumber region, the cross spectra $C_{np}^*$ also show power law type slopes. In this region, the curve of $C_{np}^*$ for $\text{St}_1 = 0.5$ and $\text{St}_2 = 0.2$ is located between the power spectra $E_{np}^*$ for $\text{St} = 0.5$ and $\text{St} = 0.2$. Similarly, the curve of $C_{np}^*$ for $\text{St}_1 = 0.5$ and $\text{St}_2 = 1.0$ is located between the power spectra $E_{np}^*$ for $\text{St} = 0.5$ and $\text{St} = 1.0$. On the other hand, in the large wavenumber region, both cross spectra $C_{np}^*$ become smaller than the power spectra $E_{np}^*$ without showing power law type slopes. These trends imply

that the cross spectrum is influenced by not only the Stokes number dependence of the clustering intensity, but also the spatial correlation of clusters between different values of St. In order to focus on the influence of the spatial correlation of clusters,

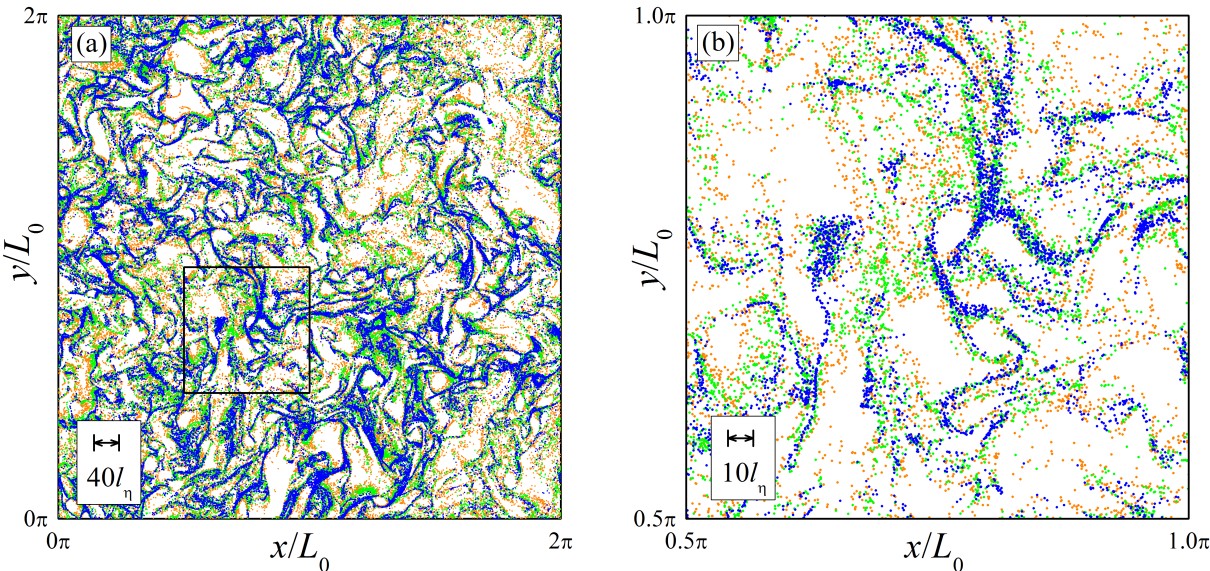

**Figure 1.** Spatial distribution of droplets for (orange) $St = 0.2$, (green) 0.5, and (blue) 1.0 in the regions of (a) $2\pi L_0 \times 2\pi L_0$ and (b) $0.5\pi L_0 \times 0.5\pi L_0$. Droplets located in the range of $0 < z < 4l_\eta$ are shown. The square frame in (a) corresponds to the region of (b).

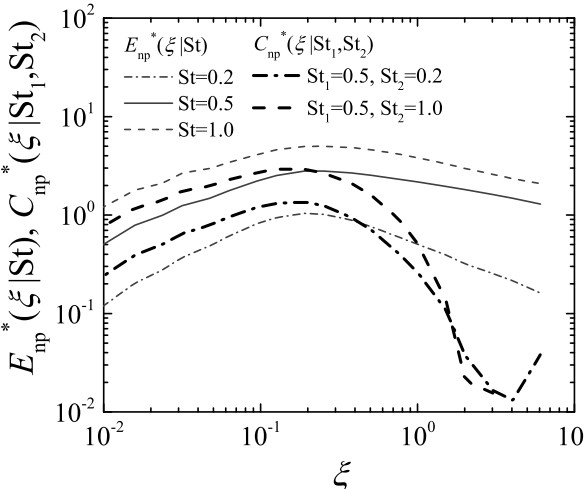

**Figure 2.** Normalized cross spectra $C_{np}^*(\xi|St_1, St_2)$ of droplet number density fluctuations compared to normalized power spectra $E_{np}^*(\xi|St)$.

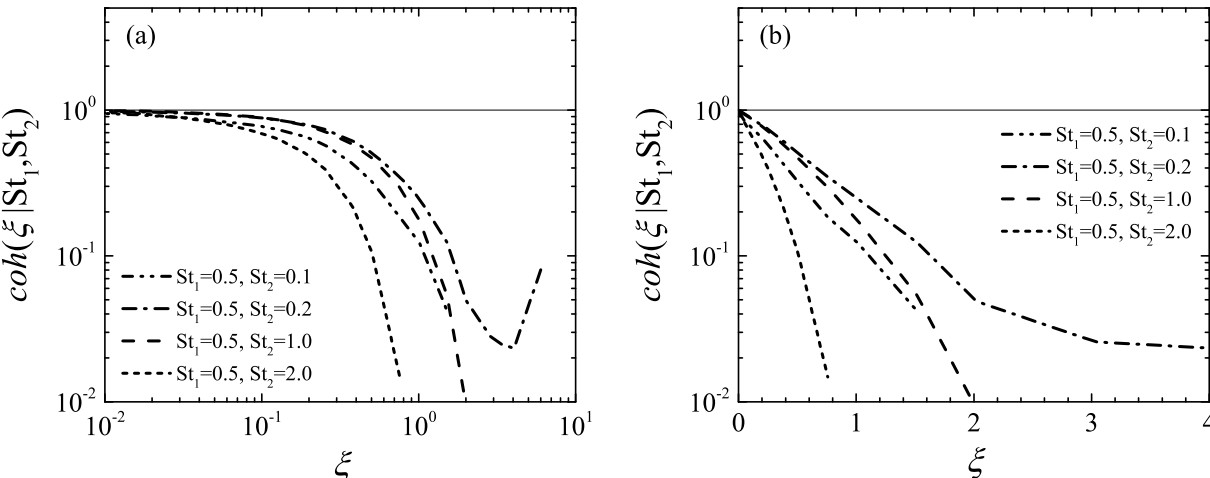

**Figure 3.** Coherence of cross spectra for combinations of $St_1 = 0.5$ and other Stokes numbers. The same coherence is plotted in (a) a double logarithmic chart and (b) a vertically logarithmic chart.

we have evaluated the coherence $coh(\xi|St_1, St_2)$, which is defined as

$$coh(\xi|St_1, St_2) = \frac{|C_{np}^*(\xi|St_1, St_2)|}{\sqrt{E_{np}^*(\xi|St_1)E_{np}^*(\xi|St_2)}}. \tag{30}$$

Figure 3 shows the coherence between $St_1 = 0.5$ and other Stokes numbers. The coherence $coh(\xi|St_1, St_2)$ is close to unity in the small wavenumber region and decreases to zero as the wavenumber increases. These trends correspond to the spatial

5  correlation between cluster locations for different $St$ cases in Figure 1. Figure 3(b) shows that the coherence decreases with an almost constant slope in the vertically logarithmic and horizontally linear chart. The slope of the coherence is dependent on the combination of $St_1$ and $St_2$; the slope becomes steeper as the difference of $St$ increases. These results indicate that the decreasing trend of coherence can be approximated by an exponential function; i.e.,

$$coh(\xi|St_1, St_2) \approx \exp\left(-\xi/\xi_c\right), \tag{31}$$

10  where $\xi_c$ is the critical wavenumber normalized by the Kolmogorov scale, given by a function of $St_1$ and $St_2$. In this study, the critical wavenumber was obtained by finding the best-fitting exponential curve to the coherence for each combination of Stokes numbers. Figure 4 shows the critical wavenumbers $\xi_c$ for all combinations of $St_1$ and $St_2$, where $St_2 > St_1$. $\xi_c$ for the same $St_1$ increases as $St_2$ decreases; whereas $\xi_c$ for the same $St_2$ increases as $St_1$ increases. This indicates that $\xi_c$ increases as $St_1$ and $St_2$ becomes closer each other. It should be noted that several studies discuss the spatial correlation of bidisperse

15  clustering particles. For example, Zhou et al. (2001) developed the radial distribution function (RDF) model at the separation length of the Kolmogorov scale. They reported that the correlation coefficient of the bidisperse RDF obtained by their DNS

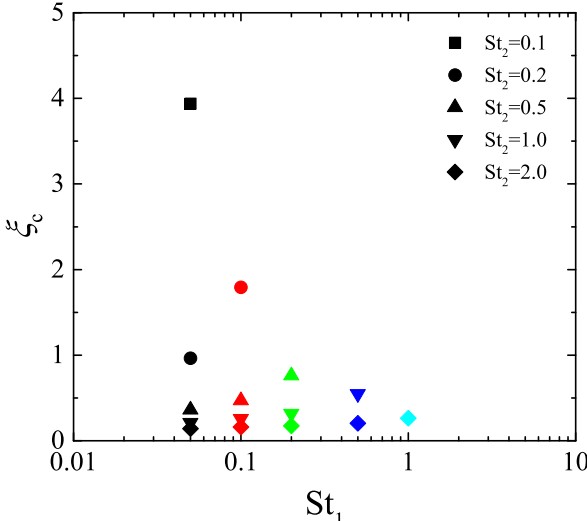

**Figure 4.** Critical wavenumber $\xi_c$ for a combination of Stokes numbers, $St_1$ and $St_2$. The symbol color and type indicate the combination of Stokes numbers. The black, red, green, blue, and light blue symbols are $St_1 = 0.05, 0.1, 0.2, 0.5$, and $1.0$, respectively. The square, circle, triangle, inverse triangle, and diamond symbols are $St_2 = 0.1, 0.2, 0.5, 1.0$, and $2.0$, respectively.

is explained well by the ratio of two Stokes numbers. Chun et al. (2005) also discussed the bidisperse RDF of clustering particles. The result of their perturbation expansion analysis indicated that the bidisperse RDF becomes constant at separation lengths sufficiently smaller than the "cross-over length," $l_c$, which is proportional to the Stokes number difference. As the cross spectrum of number density fluctuations is a Fourier transform of the bidisperse RDF, the Stokes number ratio, $St_2/St_1$, and

5 the Stokes number difference, $St_2 - St_1$, are candidates for the dominant parameter for $\xi_c$. Figure 5 shows $\xi_c$ plots against $St_2/St_1$ and $St_2 - St_1$. These figures clearly indicate that the Stokes number dependence of $\xi_c$ is explained by $St_2 - St_1$ better than $St_2/St_1$. This would be because both the critical wavenumber $\xi_c$ and cross-over length $l_c$ represent the critical scale of the spatial correlation between the clusters for different Stokes numbers; i.e., the cluster locations are less correlated at a scale smaller than the critical scale. Based on this insight, we estimate that the critical wavenumber $\xi_c$ is inversely proportional to

10 the cross-over length $l_c$. Figure 6 shows $\xi_c$ against the inverse of the Stokes number difference, which is generally expressed as $1/|St_1 - St_2|$. This figure confirms that $\xi_c$ is approximately proportional to $1/|St_1 - St_2|$; the least square fitting gives

$$\xi_c(St_1, St_2) \approx \frac{0.191}{|St_1 - St_2|}. \tag{32}$$

This implies that $\xi_c$ is closely related to the inverse of $l_c/l_\eta$ because the cross-over length of Chun et al. (2005) is approximately $l_c/l_\eta \approx 5.0|St_1 - St_2|$ based on their DNS data. It should be noted that the analytical results of Chun et al. (2005) are valid for

15 $St \ll 1$. Thus, the deviation of $\xi_c$ from the linear curve is due to the higher-order response of particle motions to the turbulent flow. However, Fig. 6 confirms that the Stokes number difference is the dominant parameter for $\xi_c$; at least for $St \leq 2.0$.

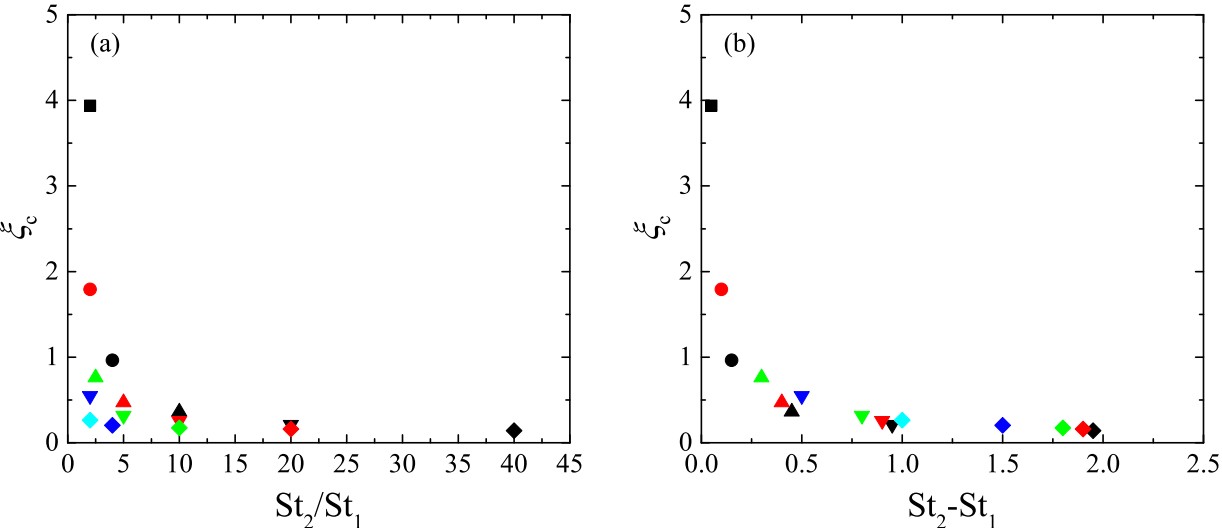

**Figure 5.** (a) Critical wavenumber $\xi_c$ against the Stokes number ratio $St_2/St_1$, and (b) against the Stokes number difference $St_2 - St_1$. Notations are as in Fig. 4.

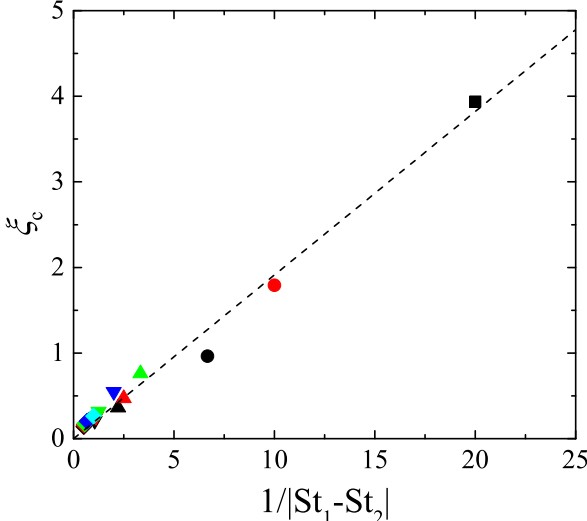

**Figure 6.** Critical wavenumber $\xi_c$ against the inverse of Stokes number difference. The dashed line is the best-fitting curve to the $\xi_c$ data. Other notations are as in Fig. 4.

## 4.2 Modeling the influence of polydisperse clustering droplets on radar reflectivity factor

According to the above discussion, we can estimate the increase in the radar reflectivity factor due to turbulent clustering of polydispere droplets in Eq. (9), provided that $q_r(r_p)$ and $\langle n_p \rangle$ are given. That is, the normalized cross spectrum $C^*_{np}(\xi|St_1, St_2)$ is given by

$$C^*_{np}(\xi|St_1, St_2) = coh(\xi|St_1, St_2)\sqrt{E^*_{np}(\xi|St_1)E^*_{np}(\xi|St_2)}. \tag{33}$$

Here, we assume that the cross spectrum is a positive real number. The coherence is estimated using Eqs. (31) and (32). The parameterization for $E^*_{np}(\xi|St)$ was proposed by Matsuda et al. (2014). The model equation is given by

$$E^*_{np}(\xi|St) = \frac{c_1 \xi^\alpha}{\left\{1 + (c_1/c_2)^{2\gamma/(\alpha-\beta)} \xi^{2\gamma}\right\}^{(\alpha-\beta)/2\gamma}}. \tag{34}$$

where $c_1$, $c_2$, $\alpha$, $\beta$, and $\gamma$ are the model parameters given by the functions of $St$ as follows:

$$\begin{cases} c_1 &= 13.4/[1 + (St/0.29)^{-1.25}], \\ c_2 &= 6.7St^{1.6}/[1 + 0.68St^{3.7}], \\ \alpha &= 0.44 - 0.20\ln St, \\ \beta &= -1 + 0.77St^{-1}\exp\left[-(\ln St - 0.55)^2/2.0\right], \\ \gamma &= 1.6. \end{cases} \tag{35}$$

Matsuda et al. (2014) confirmed that this parameterization is reliable for $St \leq 2.0$: in this range, the error of the parameterization is smaller than 1 dB.

In order to evaluate the reliability of the proposed cross spectrum model, the $r_p^3$-weighted power spectrum $E_{r3np}(k)$ for the droplet size distribution of CUMA has been estimated using the proposed cross spectrum model and compared with the spectrum obtained from the DNS data for the cases of CUMA_eps100, CUMA_eps400, and CUMA_eps1000. Figure 7 (a) shows the $r_p^3$-weighted power spectrum, which is normalized as

$$E^*_{r3np}(\xi) = \frac{E_{r3np}(k)}{\langle r_p^3 \rangle^2 \langle n_p \rangle^2 l_\eta}, \tag{36}$$

where $\langle r_p^3 \rangle$ is given by $\langle r_p^3 \rangle = \int_0^\infty r_p^3 q_r(r_p)dr_p$. The dashed lines show $E^*_{r3np}(\xi)$ predicted using the proposed parameterization including Eq. (31), while the dashed-dotted lines show those predicted by assuming perfect coherence, i.e., $coh(\xi|St_1, St_2) = 1$. The parameterization with $coh(\xi|St_1, St_2) = 1$ overestimates $E^*_{r3np}(\xi)$ at large wavenumbers and the difference becomes larger as $\epsilon$ becomes larger. This indicates that the influence of the weak spatial correlation of cluster locations between different Stokes numbers is not negligible for predicting the spectrum $E^*_{r3np}(\xi)$ for large wavenumbers, and the assumption of $coh(\xi|St_1, St_2) = 1$ can be applied only for predicting $E^*_{r3np}(\xi)$ for small wavenumbers ($\xi < O(10^{-1})$). On the other hand, $E^*_{r3np}(\xi)$ values predicted by the parameterization using Eq. (31) show good agreement with those obtained by the DNS data for overall wavenumbers. The error level of the parameterization using Eq. (31) is evaluated by the RMS error $e_{RMS}$ in units of decibels.

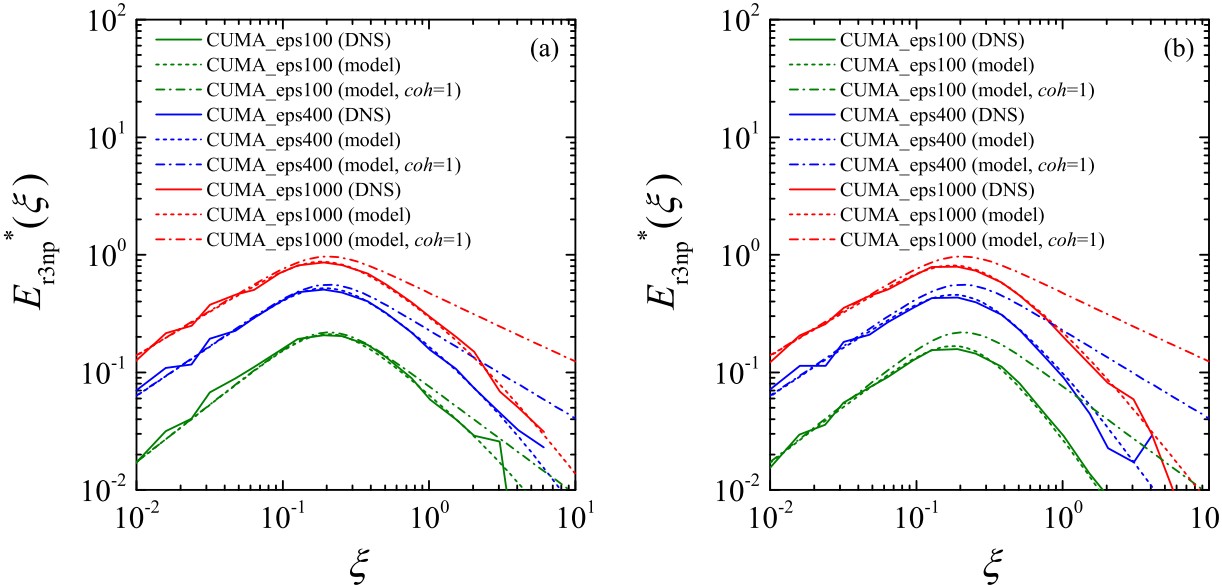

**Figure 7.** Comparisons of $r_{\text{p}}^3$-weighted power spectrum $E_{\text{r3np}}^*(\xi)$ obtained from DNS data and that estimated by the proposed cross spectrum model for the cases of (a) $g = 0.0$ and (b) 9.8 m/s$^2$.

$e_{\text{RMS}}$ is defined as

$$e_{\text{RMS}} = \frac{1}{\xi'_{\text{max}} - \xi'_{\text{min}}} \int_{\xi'_{\text{min}}}^{\xi'_{\text{max}}} \left\{ E_{\text{r3np,model}}^{*\ \text{dB}}(\xi) - E_{\text{r3np,DNS}}^{*\ \text{dB}}(\xi) \right\}^2 d\xi', \tag{37}$$

where $\xi'$ is defined as $\xi' = \ln \xi$ and superscript dB denotes a value in units of decibels. $e_{\text{RMS}}$ was calculated for the wavenumber range relevant to radar observations; i.e., $0.05 \leq \xi \leq 4.0$. $e_{\text{RMS}}$ for the cases of CUMA_eps100, CUMA_eps400, and CUMA_eps1000 are 1.41, 0.152, and 0.251 dB, respectively. Because the error level of 1 dB is unavoidable for radar observations (Bringi et al., 1990; Carey et al., 2000), $e_{\text{RMS}}$ values for CUMA_eps400 and CUMA_eps1000 are negligibly small. $e_{\text{RMS}}$ for CUMA_eps100 is slightly larger than the threshold, but this is caused by the error of calculating the reference spectrum based on the DNS data at $\xi > 2$. We confirm that, for CUMA_eps100, $e_{\text{RMS}}$ evaluated at the range of $0.05 \leq \xi \leq 2.0$ is smaller than 1 dB. Thus, the Stokes number dependence of the cross spectrum in the absence of gravity is appropriately modeled to predict the influence of turbulent clustering to a sufficient accuracy.

### 4.3 Influence of gravitational settling on cross spectrum

The parameterization summarized in the previous subsection was obtained under the condition without gravitational droplet settling. The settling influence for the monodispersed cases was discussed by Matsuda et al. (2014) and Matsuda et al. (2017).

The large gravitational settling can modulate $E_{\text{np}}^*(\xi|\text{St})$, and that can be a cause of significant difference of the radar reflectivity factor. However, the influence on $E_{\text{np}}^*(\xi|\text{St})$ is insignificant for $S_v < 3$ (Matsuda et al., 2014).

For the cases of polydisperse particles, the settling influence on the coherence term must be considered as well as the influence on $E_{\text{np}}^*(\xi|\text{St})$ in Eq. (33). Ayala et al. (2008a) and Lu et al. (2010) reported that gravitational settling enlarges the cross-over length of the bidisperse RDF. Lu et al. (2010) extended the perturbation expansion analysis of Chun et al. (2005) and presented the formulation for the cross-over length in the presence of gravity, which is

$$\frac{l_c}{l_\eta} = C_{\text{Chun}} |\text{St}_1 - \text{St}_2| \left[ 1 + \frac{1}{3a_0} \left( \frac{\tau_g}{\tau_a} \right) \text{Fr}^{-2} \right]^{1/2}, \tag{38}$$

where $C_{\text{Chun}}$ is the coefficient derived by Chun et al. (2005), $a_0$ is the ratio of the acceleration variance $\langle a^2 \rangle$ to square of the Kolmogorov acceleration $a_\eta^2$, $\tau_a$ is the acceleration correlation time scale, and $\tau_g$ is the correlation time scale for gravitational settling particles. Chun et al. (2005) obtained the values of $C_{\text{Chun}} \approx 5.0$ and $a_0 \approx 1.545$ based on their DNS results. Lu et al. (2010) further assumed $S_v \lesssim 1$ so that $\tau_g$ is simply given by $\tau_g = \tau_a = 1.5\tau_\eta$. The cross-over length $l_c$ of Eq. (38) becomes equivalent to that of Chun et al. (2005) when gravitational settling is negligibly small, whereas the gravity effect is dominant for the cases of $\text{Fr} < 0.47$, that often appears in cloud turbulence. Thus, in this study, we modify Eq. (32) to include the settling influence on the coherence $coh(\xi|\text{St}_1, \text{St}_2)$. Since $\xi_c$ is inversely proportional to $l_c/l_\eta$, we propose the following correction based on Eq. (38):

$$\xi_c(\text{St}_1, \text{St}_2) = \frac{0.191}{|\text{St}_1 - \text{St}_2|} \left[ 1 + \frac{1}{3a_0} \text{Fr}^{-2} \right]^{-1/2}. \tag{39}$$

Note that this study also adopts the value of $a_0$ obtained by Chun et al. (2005).

The reliability of the modified parameterization for the case with gravitational settling has been evaluated in the same way as the previous subsection. Figure 7 (b) shows $E_{\text{r3np}}^*(\xi)$ for the case with gravitational settling. $E_{\text{r3np}}^*(\xi)$ values at large wavenumbers are smaller than those for the case without gravitational settling, and the difference from the parameterization with $coh(\xi|\text{St}_1, \text{St}_2) = 1$ is larger, indicating that the coherence model is more important than the case without gravitational settling. It is also confirmed that $E_{\text{r3np}}^*(\xi)$ values predicted by the proposed parameterization show good agreement with those of the DNS results for the case with gravitational settling. $e_{\text{RMS}}$ evaluated at the range of $0.05 \leq \xi \leq 4.0$ are 0.93 and 0.31 dB for the cases of CUMA_eps400 and CUMA_eps1000, respectively, and that for CUMA_eps100 at the range of $0.05 \leq \xi \leq 2.0$ is 0.26 dB. Thus, $e_{\text{RMS}}$ remains smaller than 1 dB even for the case with gravitational settling. These results indicate that the proposed parameterization can predict the influence of turbulent clustering for polydisperse droplets considering the gravity effect to a sufficient accuracy. For the CUMA cases in Fig. 7 (b), $S_v$ for the modal radius is smaller than unity. For the cases of $S_v > O(1)$, the proposed parameterization would become less reliable. To improve the parameterization, it is necessary to consider an anisotropic clustering structure of settling inertial particles. Inertial particles with large settling velocity form anisotropic clusters, which are vertically elongated and horizontally confined (Bec et al., 2014; Ireland et al., 2016; Matsuda et al., 2017). When the clustering structure is anisotropic, the influence on the radar reflectivity factor theoretically depends on the direction of microwave propagation.

## 5 Application to cloud simulation data

### 5.1 Cloud simulation data

We have applied the proposed model to the high-resolution cloud-simulation data of Onishi and Takahashi (2012) to investigate the influence of turbulent clustering on radar observations. They used the Multi-Scale Simulator for the Geoenvironment (MSSG), which is a multi-scale atmosphere-ocean coupled model developed by the Japan Agency for Marine-Earth Science and Technology. The atmospheric component of MSSG (MSSG-A) solves non-hydrostatic equations and predicts three wind components, air density, and pressure, as well as water substance. Finite difference schemes are used for calculating spatial derivatives. Turbulent diffusion is calculated using the static Smagorinsky model. Onishi and Takahashi (2012) used a spectral-bin scheme for liquid water to explicitly account for the droplet size distributions. The spectral bin scheme predicts the mass distribution function $G(y)$, which is given by

$$G(y)dy = n_{\mathrm{p}} m(r_{\mathrm{p}}) q_{\mathrm{r}}(r_{\mathrm{p}}) dr_{\mathrm{p}} \tag{40}$$

where $y = \ln r_{\mathrm{p}}$, and $m(r_{\mathrm{p}})$ is the mass of droplets with a radius of $r_{\mathrm{p}}$. The mass coordinate $m$ and logarithmic coordinate $y$ are discretized as

$$m_k = 2^{1/s} m_{k-1} \tag{41}$$

$$y_k = y_{k-1} + dy \tag{42}$$

where $dy = \ln 2/(3s)$, and $s$ is a constant; $s = 1$ were used. The number of bins was 33. The representative radius of the first bin, $r_{\mathrm{p}1}$, was 3 $\mu$m; thus, the representative radius of the 33rd bin (the largest droplet class) was $r_{\mathrm{p}33} = 4.9$ mm. The prognostic variable for liquid water is the water mass content, $M_k$, which is defined as $M_k = \int_{y_{k-1/2}}^{y_{k+1/2}} G(y) dy$; i.e., 33 transport equations for $M_k$ were solved in this simulation. The activation process of cloud condensation nuclei (CCN) was considered based on the Twomey's relationship between the number of activated CCN and the saturation ratio (Twomey, 1959). The activated droplets were added to the bins using the "prescribed spectrum" method (Soong, 1974). Detail of the model configuration is described in Onishi and Takahashi (2012). The model settings and computational conditions were based on the protocol of the RICO model intercomparison project (van Zanten et al., http://www.knmi.nl/samenw/rico/). The protocol is based on the rain in cumulus over the ocean (RICO) field campaign. The domain size is $12.8 \times 12.8 \times 4.0$ km. The resolution setting of the original RICO protocol is $128 \times 128$ points in horizontal directions and 100 points in the vertical direction; i.e., $\Delta_{\mathrm{x}} = \Delta_{\mathrm{y}} = 100$ m and $\Delta_{\mathrm{z}} = 40$ m. Onishi and Takahashi (2012) performed the cloud simulation for 24 h using the original resolution setting, then continued it for an additional hour using a higher resolution setting, generating $512 \times 512$ points in horizontal directions and 200 points in the vertical direction, giving grid spacing of $\Delta_{\mathrm{x}} = \Delta_{\mathrm{y}} = 25$ m and $\Delta_{\mathrm{z}} = 20$ m. This study used the temporal slice of cloud simulation data at higher resolution.

## 5.2 Computational method for radar reflectivity factor

The radar reflectivity factor $Z$ including the influence of particulate and clear-air Bragg scatterings is given by

$$Z = Z_{\text{incoh}} + Z_{\text{PB}} + Z_{\text{CB}}, \tag{43}$$

where $Z_{\text{PB}}$ is the particulate Bragg scattering part of $Z$ in Eq. (9) and $Z_{\text{CB}}$ is an additional term reflecting clear-air Bragg scattering. In real clouds, $Z_{\text{PB}}$ is caused by droplet number density fluctuations due to turbulent droplet clustering and turbulent entrainment of environmental clear air. $E_{\text{r3np}}(k)$ for these factors is given by $E_{\text{r3np}}(k) = E_{\text{r3np}}^{\text{clust}}(k) + E_{\text{r3np}}^{\text{entr}}(k)$ (Matsuda et al., 2014), where $E_{\text{r3np}}^{\text{clust}}(k)$ and $E_{\text{r3np}}^{\text{entr}}(k)$ are the power spectra for turbulent clustering and entrainment. Note that the correlation term between the cloud water fluctuations due to these factors is negligibly small because the scales of the clustering and entrainment sources are typically separated. Thus, $Z_{\text{PB}}$ is also given by the linear combination; i.e., $Z_{\text{PB}} = Z_{\text{PBc}} + Z_{\text{PBe}}$, where $Z_{\text{PBc}} = 2^7 \pi^2 \kappa^{-2} E_{\text{r3np}}^{\text{clust}}(\kappa)$ and $Z_{\text{PBe}} = 2^7 \pi^2 \kappa^{-2} E_{\text{r3np}}^{\text{entr}}(\kappa)$.

The spectrum $E_{\text{r3np}}^{\text{clust}}(k)$ was calculated using Eq. (10) and the parameterization proposed in the previous section. To determine the Stokes number for each droplet size, the energy dissipation rate $\epsilon$ of the cloud simulation data was calculated based on the Smagorinsky model; i.e.,

$$\epsilon = (C_{\text{s}} \Delta_{\text{s}})^2 \left( 2 \widehat{s_{ij}} \widehat{s_{ij}} \right)^{3/2}, \tag{44}$$

where $C_{\text{s}}$ is the Smagorinsky constant ($C_{\text{s}} = 0.173$ in this study), $\Delta_{\text{s}}$ the representative grid spacing given by $\Delta_{\text{s}} = (\Delta_{\text{x}} \Delta_{\text{y}} \Delta_{\text{z}})^{1/3}$, and $\widehat{s_{ij}}$ the strain rate tensor, which is given by $\widehat{s_{ij}} = \frac{1}{2}(\frac{\partial \widehat{u_i}}{\partial x_j} + \frac{\partial \widehat{u_j}}{\partial x_i})$, where $\widehat{u_i}$ is the air velocity in the resolved scale.

The spectrum $E_{\text{r3np}}^{\text{entr}}(k)$ was calculated using the well-known scalar concentration spectrum. Erkelens et al. (2001) estimated this contribution based on the $-5/3$ power law in the inertial-convective range of the spectrum. In this study, the $-1$ power law in the viscous-convective range ($kl_\eta < 0.1$) is also considered since the diffusive coefficient $D_{\text{np}}$ for droplet number density is much smaller than $\nu$. $E_{\text{r3np}}^{\text{entr}}(k)$ is approximately given by (Hill, 1978)

$$\frac{E_{\text{r3np}}^{\text{entr}}(k)}{\chi_{\text{r3np}} \epsilon^{-3/4} \nu^{5/4}} = C_{\text{c}} (kl_\eta)^{-5/3} \left[ \left\{ kl_\eta (C_{\text{b}}/C_{\text{c}})^{3/2} \right\}^{2\gamma'} + 1 \right]^{1/3\gamma'}, \tag{45}$$

where $\chi_{\text{r3np}}$ is the scalar dissipation rate for $\langle r_{\text{p}}^3 \rangle n_{\text{p}}$, $C_{\text{c}}$ is the Obukhov-Corrsin constant ($C_{\text{c}} = 0.67$) (Sreenivasan, 1996; Goto and Kida, 1999), $C_{\text{b}}$ is the Batchelor constant ($C_{\text{b}} = 3.7$) (Grant et al., 1968; Goto and Kida, 1999), and $\gamma'$ is the model parameter ($\gamma' = 1.4$).

The contribution of clear-air Bragg scattering was calculated by

$$Z_{\text{CB}} = 2^5 |K|^{-2} \kappa^{-2} E_{\text{n}}(\kappa), \tag{46}$$

where $E_{\text{n}}(k)$ is the power spectrum of refractive index fluctuations. The refractive index $n_{\text{ref}}$ is given approximately by (Balsley and Gage, 1980)

$$n_{\text{ref}} = 1 + 3.73 \times 10^{-1} \frac{e}{T^2} + 7.76 \times 10^{-5} \frac{p}{T}, \tag{47}$$

where $p$ and $e$ are the atmospheric pressure and partial pressure of water vapor in hPa, and $T$ is the absolute temperature. The contribution of free electrons was omitted because it is negligibly small in the troposphere. The power spectrum $E_n(k)$ is given by the scalar concentration spectrum for $\mathrm{Pr} < 1$ (Pao, 1964), where $\mathrm{Pr} \equiv \nu/D$ ($D$ is the scalar diffusive coefficient) is the Prandtl number:

$$\frac{E_n(k)}{\chi_n \epsilon^{-3/4} \nu^{5/4}} = C_c (k l_\eta)^{-5/3} \exp\left[-1.5 \frac{C_c}{\mathrm{Pr}} (k l_\eta)^{4/3}\right], \qquad (48)$$

where $\chi_n$ is the scalar dissipation rate for $n_{\mathrm{ref}}$. In this study, the Prandtl number of refractive index was set to 0.7.

The scalar dissipation rates for $\langle r_p^3 \rangle n_p$ and $n_{\mathrm{ref}}$ were calculated in the same way. That is, the dissipation rate of an arbitrary scalar $\theta$ is given by

$$\chi_\theta = 2 \frac{\nu_t}{\mathrm{Sc}_t} \frac{\partial \widehat{\theta}}{\partial x_i} \frac{\partial \widehat{\theta}}{\partial x_i}, \qquad (49)$$

where $\nu_t$ is the eddy kinematic viscosity, $\mathrm{Sc}_t$ is the turbulent Schmidt number, and $\widehat{\theta}$ is the scalar value in the resolved scale. $\nu_t$ was calculated by using the Smagorinsky model, i.e., $\nu_t = (C_s \Delta_s)^2 (2 \widehat{s_{ij} s_{ij}})^{1/2}$, and $\mathrm{Sc}_t$ was set to 0.4 (Moin et al., 1991).

## 5.3 Results and discussion

Figure 8 (a) shows the three-dimensional visualization of liquid water. The optical thickness of each grid cell, $\tau_\Delta$, is visualized by volume rendering to mimic human-eye observations of clouds. Here, the optical thickness is defined by $\tau_\Delta = $
$Q_{\mathrm{ext}} \pi \langle r_p^2 \rangle n_p \Delta_z$, where $Q_{\mathrm{ext}}$ is the extinction efficiency for Mie scattering ($Q_{\mathrm{ext}} = 2.0$ in this study), and $\langle r_p^2 \rangle$ is gven by $\langle r_p^2 \rangle = \int_0^\infty r_p^2 q_r(r_p) dr_p$. Note that the optical transmittance of cloud volume is approximately equal to $1 - \tau_\Delta$ when $\tau_\Delta$ is sufficiently smaller than unity. Figure 8 (b) shows the isosurfaces of the energy dissipation rate $\epsilon$ and the vertical velocity $u_3$. The locations of upward flows correspond to the locations of clouds and large $\epsilon$ regions are observed around the upward flows. This indicates that strong turbulence is generated by entrainment motions due to updrafts.

This study focused on a vertical cross section that slices the cumulus cloud with the largest upward velocity. Figure 9 shows the liquid water content (LWC) in a logarithmic scale, the energy dissipation rate $\epsilon$, the radar reflectivity factor $Z^{\mathrm{dB}}$, the increase of $Z^{\mathrm{dB}}$ due to particulate Bragg scattering, and particulate Bragg scattering due to turbulent clustering $Z_{\mathrm{PBc}}^{\mathrm{dB}}$ in the cross section. The microwave frequency was set to $f_m = 2.8$ GHz, which is the representative frequency of S-band radars. The radar reflectivity factor is shown in units of decibels, which is defined as $Z^{\mathrm{dB}}$ (dBZ) $= 10 \log_{10} Z (\mathrm{mm}^6/\mathrm{m}^3)$. Large values
of $Z^{\mathrm{dB}}$ are observed inside and below the clouds. The strong echo below the clouds reflects drizzling regions, where the LWC is smaller than that inside the clouds but the strong echo returns from the drizzling droplets because $Z_{\mathrm{incoh}}$ is proportional to $\langle r_p^6 \rangle$. The radar echo at outside of the isoline of $Z_{\mathrm{incoh}}^{\mathrm{dB}} = -18$ dBZ is caused by clear-air Bragg scattering; i.e., $Z_{\mathrm{CB}}$. The radar echo layer at the height from 2.2 to 2.5 km is caused by $Z_{\mathrm{CB}}$ due to a large humidity gap in the inversion layer.

Figure 9 (c) does not show a clear sign of the mantle echo reported by Knight and Miller (1998). Knight and Miller (1998)
reported that the predominant mantle echo is observed on dry days, while it is poorly observed on the most humid day. Our result is in accord with this report as the relative humidity of the environmental air is above 80% at heights below about 2.2 km in our cloud simulation data. A possible cause of the mantle echo is particulate Bragg scattering due to turbulent entrainment

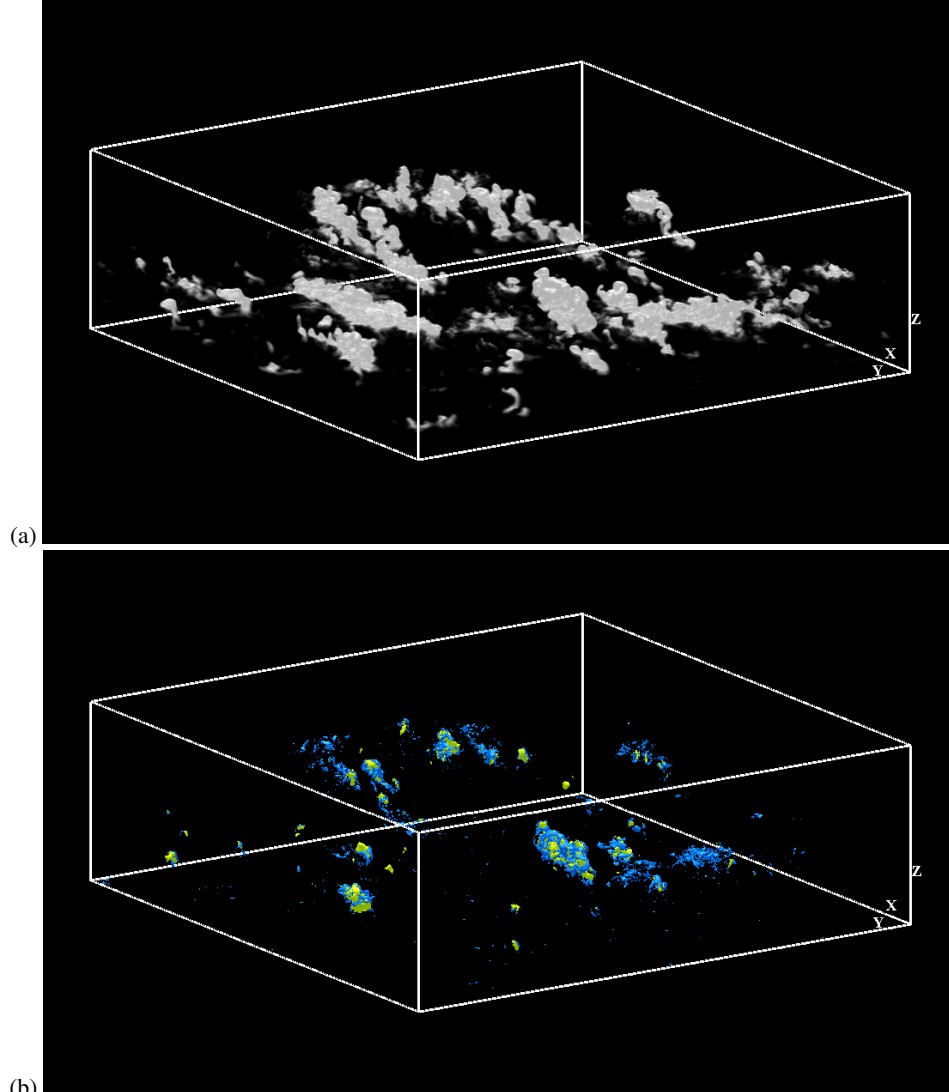

**Figure 8.** Three-dimensional visualization of cloud simulation data: (a) volume rendering of optical depth and (b) isosurfaces of (blue) the energy dissipation rate $\epsilon = 100 \text{ cm}^2/\text{s}^3$ and (yellow) the vertical velocity $u_3 = 3$ m/s.

(i.e., $Z_{\text{PBe}}$) because the large-scale cloud water inhomogeneity at cloud edges produces small-scale fluctuations due to the turbulent cascade (Erkelens et al., 2001). The influence of turbulent entrainment on $Z^{\text{dB}}$ turned out to be, however, negligibly small. That is, the fluctuations caused by the large-scale inhomogeneity were not significantly large at the scale of the half wavelength in the present simulation.

5    In Figure 9 (d), the increase due to particulate Bragg scattering, $Z^{\text{dB}} - (Z_{\text{incoh}} + Z_{\text{CB}})^{\text{dB}}$, is significant at the near-top of the clouds. The maximum difference is larger than 5 dB. In order to discuss the reason of the strong clustering influence at

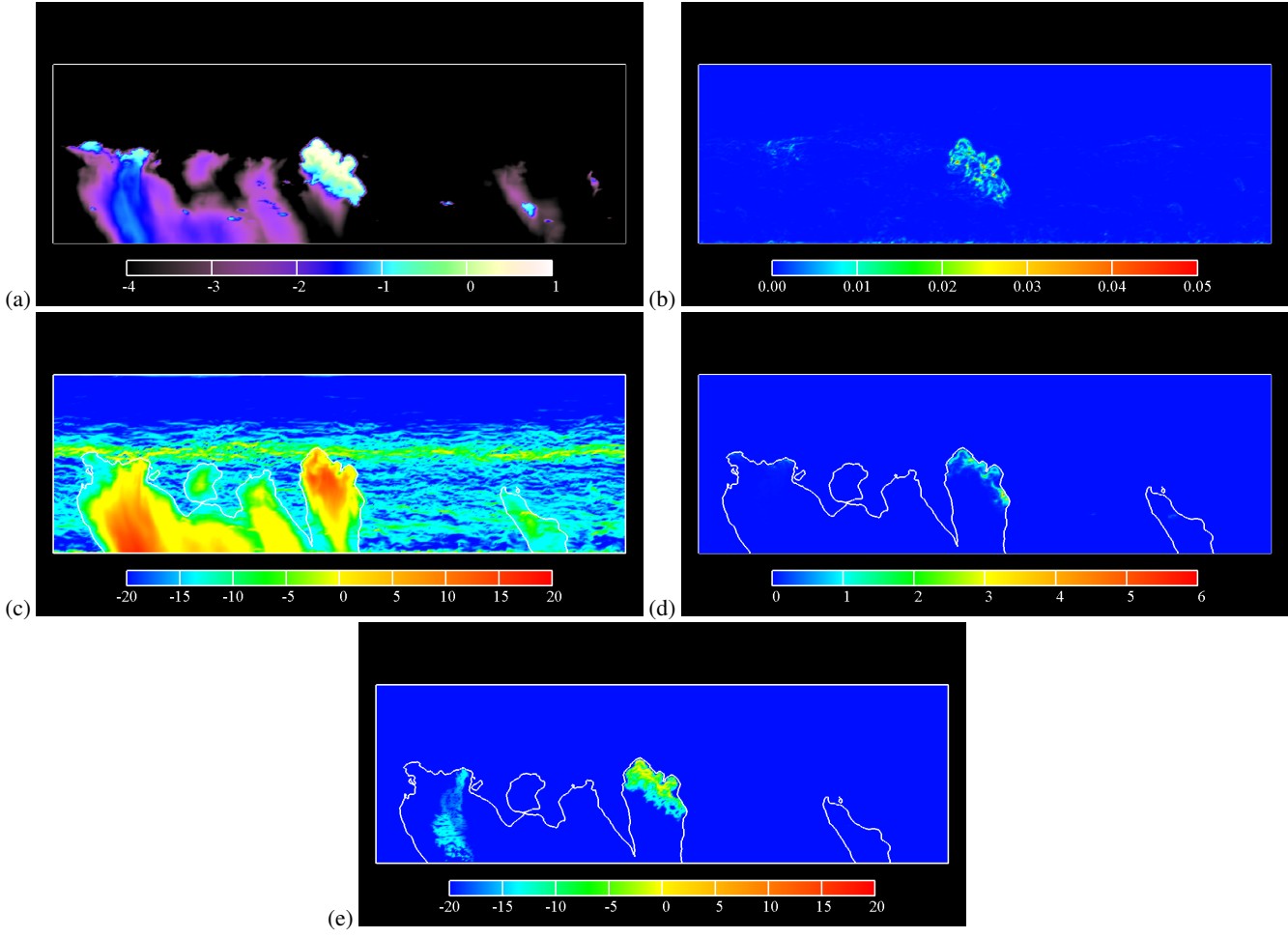

**Figure 9.** Liquid water content (LWC), energy dissipation rate $\epsilon$, and radar reflectivity factors for S-band microwaves in the vertical cross section; (a) LWC in a logarithmic scale (i.e., $\log_{10} \mathrm{LWC}$ $(\mathrm{g/m^3})$), (b) $\epsilon$ $(\mathrm{m^2/s^3})$, (c) $Z^{\mathrm{dB}}$ (dBZ), (d) $Z^{\mathrm{dB}} - (Z_{\mathrm{incoh}} + Z_{\mathrm{CB}})^{\mathrm{dB}}$ (dB), and (e) $Z_{\mathrm{PBc}}^{\mathrm{dB}}$ (dBZ). The solid lines in (c), (d), and (e) indicate the isoline of $Z_{\mathrm{incoh}}^{\mathrm{dB}} = -18$ dBZ.

the near-top of the clouds, the raw value of $Z_{\mathrm{PBc}}^{\mathrm{dB}}$ is plotted in Fig. 9 (e). $Z_{\mathrm{PBc}}^{\mathrm{dB}}$ is larger than -10 dBZ inside the turbulent cloud region, where the LWC is larger than 0.1 g/m³ and the energy dissipation rate $\epsilon$ is intermittently larger than 100 cm²/s³. Large values of $Z_{\mathrm{PBc}}^{\mathrm{dB}}$ are shown at the near-top inside this cloud region. We have confirmed that the droplet size in this cloud region was almost homogeneous: The volume-averaged droplet radius ranged within 7 to 11 $\mu$m. As a result, large values of

5  the Stokes number (up to 0.05) distributed intermittently corresponding to the distribution of $\epsilon$. The main factor of the height dependence of $Z_{\mathrm{PBc}}^{\mathrm{dB}}$ is the LWC, which is larger than 1 g/m³ at the near-top of the clouds. Note that $Z_{\mathrm{PBc}}$ is proportional to square of the LWC as Eqs. (9) and (36) implies $Z_{\mathrm{PBc}} = 2^3 3^2 \rho_{\mathrm{p}}^{-2} \kappa^{-2} (\mathrm{LWC})^2 l_\eta E_{\mathrm{r3np}}^* (\kappa l_\eta)$. Thus, the significant influence of turbulent clustering is caused by sufficiently large values of the energy dissipation rate and the LWC.

## 6 Conclusions

This study has investigated the influence of microscale turbulent clustering of polydisperse cloud droplets on the radar reflectivity factor. Firstly, the theoretical solution for particulate Bragg scattering for polydisperse droplets has been obtained considering the droplet size distribution in the measurement volume and the droplet size dependence of turbulent clustering. The obtained formula shows that the particulate Bragg scattering part of the radar reflectivity factor is given by a double integral function including the cross spectrum of number density fluctuations for bidisperse droplets. Secondly, the wavenumber and Stokes number dependence of the cross spectrum has been investigated using the turbulent droplet clustering data obtained from a direct numerical simulation (DNS) of particle-laden homogeneous isotropic turbulence without gravitational settling. The result shows that the cross spectrum for a combination of Stokes numbers, $St_1$ and $St_2$, has values between the power spectra for $St_1$ and $St_2$ at small wavenumbers, whereas the spectrum decreases more rapidly than the power spectra as the wavenumber increases. This decreasing trend is related to the scale dependence of the spatial correlation of cluster locations between two different Stokes numbers. The coherence of the cross spectrum is close to unity for small wavenumbers and decreases almost exponentially with increasing wavenumber. This is qualitatively consistent with the visualization results, in which the clustering locations for different Stokes numbers are almost the same at large scales, whereas a discrepancy in clustering locations is observed at small scales. It is also confirmed that the decreasing trend of the coherence is strongly dependent on the combination of Stokes numbers.

In order to develop a cross spectrum model for estimating the clustering influence on the radar reflectivity factor, we have proposed an exponential model for the wavenumber dependence of the coherence, and introduced the critical wavenumber (i.e., the decay constant for the model) to consider the dependence of the coherence on the Stokes number combination. The coherence data for all combinations of six Stokes numbers ranging from 0.05 to 2.0 reveals that the critical wavenumber is inversely proportional to the Stokes number difference, $|St_1 - St_2|$. This implies that the critical wavenumber is inversely proportional to the cross-over length for the bidisperse radial distribution function (RDF). The proposed coherence model enables us to estimate the cross spectrum for arbitrary combinations of Stokes numbers using the power spectrum model proposed by Matsuda et al. (2014). Comparison of the model estimate with the DNS results for a typical droplet size distribution in cumulus clouds confirms the reliability of the Stokes number dependence of the proposed model.

The proposed model has been further extended for the case with gravitational settling. We have assumed $S_v \lesssim 1$, where $S_v$ is the settling parameter, and modified the parameterization for the critical wavenumber based on the analytical equation for the cross-over length considering the settling influence (Lu et al., 2010). The $r_p^3$-weighted power spectrum estimated by the modified model shows a good agreement with that obtained by the DNS data considering the droplet size distribution in cumulus clouds and gravitational settling, indicating that the proposed model can estimate the clustering influence on the radar reflectivity factor to a sufficient accuracy.

Finally, the proposed model has been applied to high-resolution cloud-simulation data of Onishi and Takahashi (2012). The data were obtained using the Multi-Scale Simulator for the Geoenvironment (MSSG), which is a multi-scale non-hydrostatic atmosphere-ocean coupled model. The cloud and rain droplet size distribution was explicitly calculated at each grid using

a spectral-bin cloud microphysics scheme. The radar reflectivity factor has been calculated considering particulate Bragg scattering due to turbulent clustering and turbulent entrainment as well as clear-air Bragg scattering caused by temperature and humidity fluctuations. The result shows that the influence of turbulent entrainment is negligibly small in our case, whereas the influence of turbulent clustering can be significant inside turbulent clouds. The large influence is observed at the near-top of the clouds, where the liquid water content (LWC) and the energy dissipation rate are sufficiently large.

*Acknowledgements.* This research was supported by JSPS KAKENHI Grant Number JP17K14598. The numerical simulations presented were carried out on the Earth Simulator supercomputer system of the Japan Agency for Marine-Earth Science and Technology (JAMSTEC).

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
