# Peer review of "Turbulent enhancement of radar reflectivity factor for polydisperse cloud droplets"

_Atmospheric Chemistry and Physics, 2018_

## Referee Comment (RC1) · Anonymous Referee #1 · 6 Jul 2018

General evaluation: This paper reports development and application of a parameterization that describes coherent Bragg scattering of electromagnetic waves due to inertial clustering of cloud drops. This work extends the work of Matsuda et al. (JAS 2014) in two important elements: i) the formulation of the radar reflectivity is extended to include polydisperse drop size distributions, and ii) the parameterization of the clustering effect on the radar reflectivity is applied to a simulation of a cloud field of shallow precipitating cumulus clouds. Both elements are interesting and together they make the paper a useful contribution. However, the presentation requires some revisions before it is accepted. The most significant problem concerns the impact of turbulent mixing on the Bragg scattering in natural clouds and lack of its representation in the numerical simulation (the major point below). I also have a few specific comments that require

authors' attention.

Major comment:

Bragg scattering comes from fluctuation of the refractive index at scales comparable to the radar wavelength. In natural clouds (e.g., Knight and Miller JAS 1998), the Bragg scattering leads to the called "mantle echo". The mantle echo likely comes from the temperature, water vapor, and cloud water fluctuation resulting from the turbulent mixing between a cloud and its environment. I am not sure if the issue of which field fluctuations (temperature, water vapor, or cloud water) contribute most to the mantle echo is settled. But I would think that inertial droplet clustering plays insignificant role in highly inhomogeneous volumes diluted by entrainment near cloud edges that undergo turbulent stirring. In fact, because no mantle echo is simulated by the model, I feel the cloud field simulation is flawed in this respect. I feel the revised paper should include a more thorough discussion of the problem, including the missing impact of the subgrid-scale heterogeneities due to entrainment and mixing on the Bragg scattering. For the general introduction to the problem, I found the introduction to Matsuda et al. JAS 2014 paper much better.

Specific comments (more serious with *).

1. P 2. I think here the issue of what causes Bragg scattering should be introduced and discussed.

2*. I consider the omission of the gravitational acceleration in DNS simulations a serious problem. There is an extensive discussion in the literature to what extent droplet sedimentation is important for the clustering problem in natural clouds starting with Grabowski and Vaillencourt (JAS 1999). I do not think one can dismiss the impact of gravity that easily. In fact, the volumes where I expect clustering to be more important than turbulent mixing (i.e., weakly diluted cloudy volumes away from cloud edges) should feature small dissipation rates (in contrast to what line 4 on p. 6 says). Weak turbulence makes the sedimentation more important.

[Figure]

3. The smallest Stokes number considered in DNS simulation (0.05) is probably still too large for small cloud droplets and low dissipation rates.

4. I only skimmed over theoretical sections of the paper and have no comments on them.

5. P. 15. I think explaining how cloud droplets are activated would be useful. The RICO case description only states CCN concentration should be taken as 100 per cc, but no details about the activation are provided. Please add.

6*. I do not like how the eddy dissipation rate in (39) is prescribed. To me, including "resolved" and "sugbgrid-scale" contributions does not make sense. If you do not agree, please provide a reference to a previous study or a textbook that used such an approach. I assume that the model has a parameterization of unresolved turbulent transport, correct? Then this can be used to derived epsilon. Please see how others have done that, for instance, Seifert et al. for a simple Smagorinsky scheme or Wyszogrodzki et al. for a TKE scheme. Also, what is niu in (40).

7. P. 15 L. 26: Please define optical thickness.

8*. I find the discussion of RICO simulations superficial. Fig. 8 is interesting, but its discussion should be expanded. I think it would help if the LWC is plotted using a different color scale or the log scale so the extent of a cloud is shown. I have already mentioned that the simulation does not show the mantle echo. My explanation is that the simulated Bragg echo includes only droplet clustering contribution. I do not feel this is realistic considering entrainment and mixing as in my view this is the main reason for the mantle echo. Moreover, by design, the model assumes homogeneous mixing for the cloud microphysics (i.e., parameterized subgrid-scale transport and numerical diffusion are followed by an immediate homogenization of the grid volume). This is clearly unrealistic for the scales the model is able to resolve. As for the reason for the strongest simulated Bragg echo being located near the cloud top, there are two effects. First, TKE typically increase with height in shallow convection. Second, droplet size

increases with height in weakly diluted volumes as well. These two work together to increase droplet clustering and lead to the largest Bragg contribution near the cloud top. Perhaps it would be interesting to know which effect is more important: the increase of droplet size or the increase of turbulence. I think all these need to be discussed emphasizing model limitation, that is, exclusion of subgrid-scale contribution to the Bragg echo.

9. Appendix A is short and should be included in the main text.

References:

Seifert, A., L. Nuijens, and B. Stevens: Turbulence effects on warm-rain autoconversion in precipitating shallow convections, Q. J. Roy. Meteor. Soc., 136, 1753–1762

Wyszogrodzki, A. A., W. W. Grabowski, L.-P. Wang, and O. Ayala, 2013: Turbulent collision-coalescence in maritime shallow convection. Atmos. Chem. Phys., 13, 8471-8487.

---

## Referee Comment (RC2) · Anonymous Referee #2 · 15 Jul 2018

This paper extends the prior work carried out by Matsuda, Onishi and colleagues on radar Bragg scattering resulting from spatial correlations in hydrometeors in turbulent flow. Here the work is extended to polydisperse particle size distributions, which is of importance for atmospheric clouds in which size distributions are almost always broad. Specifically, it is known that polydispersity in particle Stokes number directly influences the distribution-averaged radial distribution function, and indeed, here it is shown that the Bragg scattering is also modified. The problem is of relevance to fully quantifying radar returns from clouds, and possibly in the future can lead to methods for remotely determining additional cloud properties, such as turbulence intensity or perhaps even size distribution width.

The work is sound, and would be publishable except for one critical factor that has

been neglected. The authors have used direct numerical simulations of turbulence with Lagrangian particles having finite inertia (Stokes number), but have neglected the gravitational sedimentation term. This is justified by referring to a 2017 paper in which the authors showed that the gravitational sedimentation effect is small (< 1 dB for the size and turbulence ranges of interest). The 2017 paper, however, deals only with monodisperse droplet populations, and it is reasonably well established that a key signature of gravitational settling for low-Stokes-number particle clustering is reduction of the bi-disperse radial distribution function. The effect is similar to the bi-disperse clustering effect described in the theoretical treatment of Chun et al. (2005, see their section 2.3). It leads to saturation of the power-law dependence of the bi-disperse radial distribution function. This cross-over scale is discussed by the authors in section 4.1, but the role of gravity in modifying that is not discussed. The radial distribution function is directly and quantitatively connected to the Bragg signature, so it is very likely that the sedimentation-induced saturation effect will be a first-order effect. The saturation effect resulting from particle settling has been confirmed in subsequent DNS and experimental studies (e.g., Ireland et al. "The effect of Reynolds number on inertial particle dynamics in isotropic turbulence. Part 2. Simulations with gravitational effects" J. Fluid Mechanics 2016, see their section 4.2; Lu et al. "Clustering of settling charged particles in turbulence: theory and experiments" New J. Physics 2010, see their section 4).

The current neglect of sedimentation requires, at least, major revision of the manuscript. Simulations that include gravitational sedimentation should be used, and perhaps compared to those without sedimentation. If the results are as strongly influenced by the gravitational settling effect as I suspect they will be, then it will be necessary to rewrite the results and conclusions sections of the paper.

---

## Author Response (AR1)

Reply to Referee #1

Thank you for your encouraging and insightful comments on our manuscript. We have intensely revised our manuscript according to your comments, particularly to the four major criticisms. The revised manuscript now includes the discussion of the mantle echo and considers the influence of gravitational acceleration. Below we answer all your comments one by one.

**[Major criticism 1]**
*Bragg scattering comes from fluctuation of the refractive index at scales comparable to the radar wavelength. In natural clouds (e.g., Knight and Miller JAS 1998), the Bragg scattering leads to the called "mantle echo". The mantle echo likely comes from the temperature, water vapor, and cloud water fluctuation resulting from the turbulent mixing between a cloud and its environment. I am not sure if the issue of which field fluctuations (temperature, water vapor, or cloud water) contribute most to the mantle echo is settled. But I would think that inertial droplet clustering plays insignificant role in highly inhomogeneous volumes diluted by entrainment near cloud edges that undergo turbulent stirring. In fact, because no mantle echo is simulated by the model, I feel the cloud field simulation is flawed in this respect. I feel the revised paper should include a more thorough discussion of the problem, including the missing impact of the subgrid-scale heterogeneities due to entrainment and mixing on the Bragg scattering. For the general introduction to the problem, I found the introduction to Matsuda et al. JAS 2014 paper much better.*

According to this criticism, we have extensively modified the manuscript in order to include the discussion on the "mantle echo". That is, we have incorporated the influences of particulate Bragg scattering due to turbulent entrainment of environmental air into clouds and clear-air Bragg scattering, which is caused by the refractive index fluctuations (i.e., temperature and humidity fluctuations). We have confirmed that the influence of turbulent clustering is still significant even if the mantle echo physics (i.e., particulate and clear-air Brag scatterings) is included. The detailed calculation method of the two kinds of Bragg scatterings has been described in Subsection 5.2 and the results in Fig. 9 have been updated.

As described in detail in Subsection 5.2 in the revised manuscript, particulate and clear-air Bragg scatterings are considered for the calculation of the radar reflectivity factor $Z$. The contributions are denoted by $Z_{PB}$ and $Z_{CB}$. For the calsulation of the entrainment contribution to $Z_{PB}$ (denoted by $Z_{PBe}$) and $Z_{CB}$, the -5/3 power law of the scalar concentration spectrum is assumed for the inertial-convective range. Figure 9 (c) now shows the results of $Z$ considering the above $Z_{PBe}$ and $Z_{CB}$ in addition to the clustering contribution to $Z_{PB}$ (denoted by $Z_{PBc}$). The newly-simulated radar echo (Fig. 9 (c)) shows a radar echo layer at the height from 2.2 to 2.5 km. This echo layer is caused by clear-air Bragg scattering due to a large humidity gap in the inversion layer. The echo figure, however, does not show a clear mantle echo. Knight and Miller (1998) reported that the mantle echo is poorly observed on the most humid day. Our result is in accord with this report as the relative humidity of the environmental air is above 80% at heights below about 2.2 km in our cloud simulation data. As you pointed out, particulate Bragg scattering due to turbulent entrainment is a possible cause of the mantle echo. The influence of turbulent entrainment turned out to be, however, not significant in our simulation results. Our interpretation is that the large-scale cloud water inhomogeneity

at cloud edges could produce small-scale fluctuations due to the turbulent cascade, but such fluctuations were not significantly large at the scale of the half wavelength in the present simulation.

According to the comment on Introduction, we have added the introduction of clear-air Bragg scattering and discussion about possible causes of the Bragg scattering observed by Knight and Miller (1998) in the revised manuscript.

**[Minor criticism 1]**

*1. P 2. I think here the issue of what causes Bragg scattering should be introduced and discussed.*

Accordingly, in Introduction, we have added the introduction of clear-air Bragg scattering and discussion about possible causes of the Bragg scattering observed by Knight and Miller (1998) in the revised manuscript.

**[Major criticism 2]**

*2\*. I consider the omission of the gravitational acceleration in DNS simulations a serious problem. There is an extensive discussion in the literature to what extent droplet sedimentation is important for the clustering problem in natural clouds starting with Grabowski and Vaillencourt (JAS 1999). I do not think one can dismiss the impact of gravity that easily. In fact, the volumes where I expect clustering to be more important than turbulent mixing (i.e., weakly diluted cloudy volumes away from cloud edges) should feature small dissipation rates (in contrast to what line 4 on p. 6 says). Weak turbulence makes the sedimentation more important.*

The other referee also criticized about the omission of the gravitational acceleration. We therefore decided to include it in the revised manuscript. The modeling of the influence of gravitational settling is described in Subsection 4.3 and the reliability of the model is confirmed in Fig. 7 (b). The radar echo simulation in Fig. 9 has been obtained with the updated model that considers the gravitational settling. It has confirmed that the influence of turbulent clustering is still significant even if the gravitational settling is considered.

In our previous work (Matsuda et al., 2014), we confirmed that the settling influence on the power spectrum for monodisperse droplets is insignificant for $S_\mathrm{v} < 3$, where $S_\mathrm{v}$ is the settling parameter defined by the ratio of the terminal velocity to the Kolmogorov velocity. However, it is true that the gravitational settling modifies the cross-over length of the radial distribution function (RDF) for bidisperse particles. Lu et al. (2010) analytically proposed the cross-over length for gravitational settling bidisperse particles. Following this analysis, we have modified our parameterization for the critical wavenumber $\xi_\mathrm{c}$ in Eq. (32) to consider the settling influence on the coherence. Since $\xi_\mathrm{c}$ is inversely proportional to the cross-over length, we propose the following correction based on the equation of Lu et al. (2010):

$$\xi_\mathrm{c} = \frac{0.191}{|\mathrm{St}_1 - \mathrm{St}_2|}\left[1 + \frac{1}{3a_0}\mathrm{Fr}^{-2}\right]^{-1/2}.$$

In order to confirm the reliability of the parametrization, we have performed additional DNSs for polydisperse droplets considering the gravitational settling. As shown in Fig. 7 (b), $E^*_{\mathrm{r3np}}(\xi)$ values obtained by the additional DNSs are smaller than those for the case without gravitational settling at large wavenumbers, indicating

that the coherence model is more important than the case without gravitational settling. $E^*_{\text{r3np}}(\xi)$ values predicted by our modified parameterization show good agreement with those of the DNS results with gravitational settling. We have also confirmed that the RMS error $e_{\text{RMS}}$ evaluated by Eq. (39) remains smaller than 1 dB even for the case with gravitational settling. These results indicate that the proposed parameterization can predict the influence of turbulent clustering for polydisperse droplets considering the gravity effect within 1 dB error.

We have added Subsection 4.3 to explain the modeling of the influence of gravitational settling. We have also modified the computational conditions accordingly in Subsection 3.1. The clustering influence in Figure 9 is also updated using the parameterization considering the gravitational settling influence. The results confirm that the influence of turbulent clustering is still significant even if the gravitational acceleration is considered.

**[Minor criticism 2]**

*3. The smallest Stokes number considered in DNS simulation (0.05) is probably still too large for small cloud droplets and low dissipation rates.*

Both in Fig. 7 (a) and (b), the spectra obtained by the proposed model show good agreement with those obtained by the DNS for the case of CUMA_eps100, where the Stokes number for the modal radius was as small as 0.035 and the energy dissipation was as small as 100 cm$^2$/s$^3$. Thus, our parameterizations for the clustering influence for both gravity and non-gravity cases still work for particles with St < 0.05. It should be noted that the model prediction is less reliable for St << 0.05 but the influence of clustering on the radar reflectivity factor for such small droplets becomes negligibly small as well.

**[Comment 1]**

*4. I only skimmed over theoretical sections of the paper and have no comments on them.*

We appreciate your attention to the theoretical part.

**[Minor criticism 3]**

*5. P. 15. I think explaining how cloud droplets are activated would be useful. The RICO case description only states CCN concentration should be taken as 100 per cc, but no details about the activation are provided. Please add.*

Accordingly, we have added the explanation about the activation model in Subsection 5.1 in the revised manuscript as follows:
"The activation process of cloud condensation nuclei (CCN) was considered based on the Twomey's relationship between the number of activated CCN and the saturation ratio (Twomey, 1959). The activated droplets were added to the bins using the ``prescribed spectrum'' method (Soong, 1974). Detail of the model configuration is described in Onishi and Takahashi (2012)."

**[Major criticism 3]**

*6\*. I do not like how the eddy dissipation rate in (39) is prescribed. To me, including "resolved" and "sugbgrid-scale" contributions does not make sense. If you do not agree, please provide a reference to a previous study or a textbook that used such an approach. I assume that the model has a parameterization of unresolved turbulent transport, correct? Then this can be used to derived epsilon. Please see how others have done that, for instance, Seifert et al. for a simple Smagorinsky scheme or Wyszogrodzki et al. for a TKE scheme. Also, what is niu in (40).*

We have removed the "resolved scale" contribution to the dissipation rate as it is indeed negligibly small compared to the "subgrid scale" contribution in the present LES. The corresponding equation, Eq. (44), and description have been modified accordingly.

Wyszogrodzki et al. (2013) used a TKE-based scheme to calculate the epsilon, and actually Seifert et al. (2010) used the same scheme to obtain the epsilon (Seifert et al. used the Smagorinsky scheme to obtain the TKE). Our scheme is based not on the TKE but on the velocity gradient tensor in the resolved scale. The epsilon in this scheme is directly derived from the LES filtering and the Smagorinsky eddy viscosity, and the contribution of the resolved scale is given separately using the molecular kinematic viscosity, which is $\nu$ in Eq. (40) in the previous manuscript.

**[Minor criticism 4]**

*7. P. 15 L. 26: Please define optical thickness.*

We have added the definition of the optical thickness in Subsection 5.3 in the revised manuscript:

"The optical thickness of each grid cell, $\tau_\Delta$, is visualized by volume rendering to mimic human-eye observations of clouds. Here, the optical thickness is defined by $\tau_\Delta = Q_{\text{ext}} \pi \langle r_p^2 \rangle n_p \Delta_z$, where $Q_{\text{ext}}$ is the extinction efficiency for Mie scattering ($Q_{\text{ext}} = 2.0$ in this study), and $\langle r_p^2 \rangle = \int_0^\infty r_p^2 q_r(r_p) dr_p$. Note that the optical transmittance of cloud volume is approximately equal to $1 - \tau_\Delta$ when $\tau_\Delta$ is sufficiently smaller than unity."

**[Major criticism 4]**

*8\*. I find the discussion of RICO simulations superficial. Fig. 8 is interesting, but its discussion should be expanded.*

*(1) I think it would help if the LWC is plotted using a different color scale or the log scale so the extent of a cloud is shown.*

*(2) I have already mentioned that the simulation does not show the mantle echo. My explanation is that the simulated Bragg echo includes only droplet clustering contribution. I do not feel this is realistic considering entrainment and mixing as in my view this is the main reason for the mantle echo. Moreover, by design, the model assumes homogeneous mixing for the cloud microphysics (i.e., parameterized subgrid-scale transport and numerical diffusion are followed by an immediate homogenization of the grid volume). This is clearly unrealistic for the scales the model is able to resolve.*

*(3) As for the reason for the strongest simulated Bragg echo being located near the cloud top, there are two effects. First, TKE typically increase with height in shallow convection. Second, droplet size increases with height in weakly diluted volumes as well. These two work together to increase droplet clustering and lead to the largest*

*Bragg contribution near the cloud top. Perhaps it would be interesting to know which effect is more important: the increase of droplet size or the increase of turbulence.*

*I think all these need to be discussed emphasizing model limitation, that is, exclusion of subgrid-scale contribution to the Bragg echo.*

(1) Accordingly, the LWC distribution in Fig. 9 (a) has been replaced to the log scale to show the extent of clouds and precipitation.

(2) As mentioned above (Major criticism 1), we have incorporated the particulate Bragg scattering due to turbulent entrainment of cloud volume with clear air and the clear-air Bragg scattering. We have added the discussion on the mantle echo and the influence of cloud water inhomogeneity in Subsection 5.3.

(3) We have added the discussion on the height dependence of the clustering influence accordingly. The strong influence of turbulent clustering is caused not only by the energy dissipation rate (the contribution of turbulent intensity) and droplets size but also by the LWC. We have added Figure 9 (e), which shows the raw value of particulate Bragg scattering attributed to turbulent clustering, $Z_{\mathrm{PBc}}^{\mathrm{dB}}$, and explained the possible reason at the last paragraph of Subsection 5.3 in the revised manuscript as:

"In order to discuss the reason of the strong clustering influence at the near-top of the clouds, the raw value of $Z_{\mathrm{PBc}}^{\mathrm{dB}}$ is plotted in Fig. 9 (e). $Z_{\mathrm{PBc}}^{\mathrm{dB}}$ is larger than -10 dBZ inside the turbulent cloud region, where the LWC is larger than 0.1 g/m$^3$ and the energy dissipation rate $\epsilon$ is intermittently larger than 100 cm$^2$/s$^3$. Large values of $Z_{\mathrm{PBc}}^{\mathrm{dB}}$ are shown at the near-top inside this cloud region. We have confirmed that the droplet size in this cloud region was almost homogeneous: The volume-averaged droplet radius ranged within 7 to 11 μm. As a result, large values of the Stokes number (up to 0.05) distributed intermittently corresponding to the distribution of $\epsilon$. The main factor of the height dependence of $Z_{\mathrm{PBc}}^{\mathrm{dB}}$ is the LWC, which is larger than 1 g/m$^3$ at the near-top of the clouds. Note that $Z_{\mathrm{PBc}}$ is proportional to square of the LWC as Eqs. (9) and (36) implies $Z_{\mathrm{PBc}} = 2^3 3^2 \rho_{\mathrm{p}}^{-2} \kappa^{-2} (\mathrm{LWC})^2 l_\eta E_{\mathrm{r3np}}^* (\kappa l_\eta)$. Thus, the significant influence of turbulent clustering is caused by sufficiently large values of the energy dissipation rate and the LWC."

**[Minor criticism 5]**

*9. Appendix A is short and should be included in the main text.*

Accordingly, we have moved the description in Appendix A to the end of the first paragraph of Subsection 4.2 in the main text.

Reply to Referee #2

We greatly appreciate your insightful comment on our paper. We have revised our results and manuscript according to your comment. The revised manuscript now considers the influence of gravitational acceleration. Below we answer your comment.

**[Referee's criticism]**

*The authors have used direct numerical simulations of turbulence with Lagrangian particles having finite inertia (Stokes number), but have neglected the gravitational sedimentation term. This is justified by referring to a 2017 paper in which the authors showed that the gravitational sedimentation effect is small (< 1 dB for the size and turbulence ranges of interest). The 2017 paper, however, deals only with monodisperse droplet populations, and it is reasonably well established that a key signature of gravitational settling for low-Stokes-number particle clustering is reduction of the bi-disperse radial distribution function. The effect is similar to the bi-disperse clustering effect described in the theoretical treatment of Chun et al. (2005, see their section 2.3). It leads to saturation of the power-law dependence of the bi-disperse radial distribution function. This cross-over scale is discussed by the authors in section 4.1, but the role of gravity in modifying that is not discussed. The radial distribution function is directly and quantitatively connected to the Bragg signature, so it is very likely that the sedimentation-induced saturation effect will be a first-order effect. The saturation effect resulting from particle settling has been confirmed in subsequent DNS and experimental studies (e.g., Ireland et al. "The effect of Reynolds number on inertial particle dynamics in isotropic turbulence. Part 2. Simulations with gravitational effects" J. Fluid Mechanics 2016, see their section 4.2; Lu et al. "Clustering of settling charged particles in turbulence: theory and experiments" New J. Physics 2010, see their section 4).*

We agree that gravitational settling must be considered for reliable modeling of turbulent clustering of cloud droplets. We had expected that numerous DNS experiments are necessary to parameterize the settling influence, but we have succeeded in incorporating the gravitational setting influence into our parameterization since you introduced the useful papers to us. The modeling of the influence of gravitational settling is described in Subsection 4.3 and the reliability of the model is confirmed in Fig. 7 (b). The radar echo simulation in Fig. 9 has been obtained with the updated model that considers the gravitational settling. It has confirmed that the influence of turbulent clustering is still significant even if the gravitational settling is considered.

In our previous work (Matsuda et al., 2014), we confirmed that the influence on the power spectrum for monodispersed droplets is insignificant for $S_v < 3$, where $S_v$ is the settling parameter defined by the ratio of the terminal velocity to the Kolmogorov velocity. However, it is true that gravitational settling modifies the cross-over length of the bidisperse RDF. Lu et al. (2010) analytically proposed the cross-over length for gravitational settling bidisperse particles. Following this analysis, we have modified our parameterization for the critical wavenumber $\xi_c$ in Eq. (32) to consider the settling influence on the coherence. Since $\xi_c$ is inversely proportional to the cross-over length, we propose the following correction based on the equation of Lu et al. (2010):

$$\xi_c = \frac{0.191}{|\mathrm{St}_1 - \mathrm{St}_2|}\left[1 + \frac{1}{3a_0}\mathrm{Fr}^{-2}\right]^{-1/2}.$$

In order to confirm the reliability of the parametrization, we have performed additional DNSs for polydisperse droplets considering the gravitational settling. As shown in Fig. 7 (b), $E^*_{\mathrm{r3np}}(\xi)$ values obtained by the additional DNSs are smaller than those for the case without gravitational settling at large wavenumbers, indicating that the coherence model is more important than the case without gravitational settling. $E^*_{\mathrm{r3np}}(\xi)$ values predicted by the modified parameterization show good agreement with those of the DNS results with gravitational settling. We have also confirmed that the RMS error $e_{\mathrm{RMS}}$ evaluated by Eq. (39) remains smaller than 1 dB even for the case with gravitational settling. These results indicate that the modified parameterization can predict the influence of turbulent clustering for polydisperse droplets considering the gravity effect within 1 dB error.

We have added Subsection 4.3 to explain the modeling of the gravitational settling influence. We have also modified the computational condition in Subsection 3.1. The clustering influence in Figure 9 is also updated using the parameterization considering the gravitational settling influence. The results confirm that the influence of turbulent clustering is still significant even if the gravitational acceleration is considered.

[revised manuscript text omitted]
_p$, $n_p(\mathbf{x}|r_p)$. The PDF is defined as $q_r(r_p) \equiv \frac{1}{N_p} \int_{\mathbf{x} \in V} \overline{n(\mathbf{x}, r_p)} d\mathbf{x}$, where $N_p$ is the total number of droplets in the measurement volume; i.e., $N_p \equiv \int_0^\infty \int_{\mathbf{x} \in V} \overline{n(\mathbf{x}, r_p)} d\mathbf{x} dr_p$. The PDF satisfies $\int_0^\infty q_r(r_p) dr_p = 1$. The number density distribution function for monodisperse droplets is then defined as $n_p(\mathbf{x}|r_p) \equiv n(\mathbf{x}, r_p)/q_r(r_p)$ so that $n(\mathbf{x}, r_p)$ is given by $n(\mathbf{x}, r_p) = n_p(\mathbf{x}|r_p) q_r(r_p)$. The number density distribution function $n_p(\mathbf{x}|r_p)$ satisfies $\int_{\mathbf{x} \in V} \overline{n_p(\mathbf{x}|r_p)} d\mathbf{x} = N_p$ for arbitrary $r_p$. Note that the spatial correlation function $\langle \delta n(\mathbf{x}, r_p) \delta n(\mathbf{x} + \mathbf{r}, r_p') \rangle$ for $r_p' = r_p$ is discontinuous at $\mathbf{r} = \mathbf{0}$ because the

25   droplet distribution is composed of spatially discrete points. The singularity is given by $\langle n(\mathbf{x}, r_p) \rangle \delta(\mathbf{r}) \delta(r_p' - r_p)$, where $\delta(\mathbf{r})$ and $\delta(r_p' - r_p)$ are the Dirac delta functions. Thus, the spatial correlation function is given by

$$\langle \delta n(\mathbf{x}, r_p) \delta n(\mathbf{x} + \mathbf{r}, r_p') \rangle = \langle n_p \rangle \delta(\mathbf{r}) q_r(r_p) \delta(r_p' - r_p) + \Psi(\mathbf{r}|r_p, r_p') q_r(r_p) q_r(r_p'), \tag{8}$$

where $\langle n_p \rangle$ is the averaged number density ($\langle n_p \rangle \equiv N_p/V$) and $\Psi(\mathbf{r}|r_p, r_p')$ is defined as the continuous part of $\langle \delta n_p(\mathbf{x}|r_p) \delta n_p(\mathbf{x} + \mathbf{r}|r_p') \rangle$. Substitution of Eq. (8) into Eq. (7) and adoption of the isotropic clustering assumption (Gossard and Strauch, 1983) yield

$$Z = 2^6 \langle r_p^6 \rangle \langle n_p \rangle + 2^7 \pi^2 \kappa^{-2} E_{\mathrm{r3np}}(\kappa), \tag{9}$$

5  where $\kappa$ is $\kappa = |\boldsymbol{\kappa}|$, $\langle r_p^6 \rangle$ is given by $\langle r_p^6 \rangle = \int_0^\infty r_p^6 q_r(r_p) dr_p$, and $E_{\mathrm{r3np}}(k)$ is the $r_p^3$-weighted power spectrum, defined as

$$E_{\mathrm{r3np}}(k) \equiv \int_0^\infty \int_0^\infty r_p^3 r_p'^3 q_r(r_p) q_r(r_p') C_{\mathrm{np}}(k|r_p, r_p') dr_p dr_
[revised manuscript text omitted]

$$e_{\mathrm{RMS}} = \frac{1}{\xi'_{\mathrm{max}} - \xi'_{\mathrm{min}}} \int_{\xi'_{\mathrm{min}}}^{\xi'_{\mathrm{max}}} \left\{ E_{\mathrm{r3np,model}}^{*\,\mathrm{dB}}(\xi) - E_{\mathrm{r3np,DNS}}^{*\,\mathrm{dB}}(\xi) \right\}^2 d\xi', \tag{37}$$

where $\xi'$ is defined as $\xi' = \ln \xi$ and superscript dB denotes a value in units of decibels. $e_{\mathrm{RMS}}$ was calculated for the wavenumber range relevant to radar observations; i.e., $0.05 \leq \xi \leq 4.0$. $e_{\mathrm{RMS}}$ for the cases of CUMA_eps100, CUMA_eps400, and CUMA_eps1000 are 1.41, 0.152, and 0.251 dB, respectively. Because the error level of 1 dB is unavoidable for radar observations (Bringi et al., 1990; Carey et al., 2000), $e_{\mathrm{RMS}}$ values for CUMA_eps400 and CUMA_eps1000 are negligibly small. $e_{\mathrm{RMS}}$ for CUMA_eps100 is slightly larger than the threshold, but this is caused by the error of calculating the reference spectrum based on the DNS data at $\xi > 2$. We confirm that, for CUMA_eps100, $e_{\mathrm{RMS}}$ evaluated at the range of $0.05 \leq \xi \leq 2.0$ is smaller than 1 dB. Thus, the Stokes number dependence of the cross spectrum in the absence of gravity is appropriately modeled to predict the influence of turbulent clustering to a sufficient accuracy.

**4.3 Influence of gravitational settling on cross spectrum**

The parameterization summarized in the previous subsection was obtained under the condition without gravitational droplet settling. The settling influence for the monodispersed cases was discussed by Matsuda et al. (2014) and Matsuda et al. (2017).

The large gravitational settling can modulate $E^*_{\mathrm{np}}(\xi|\mathrm{St})$, and that can be a cause of significant difference of the radar reflectivity factor. However, the influence on $E^*_{\mathrm{np}}(\xi|\mathrm{St})$ is insignificant for $S_{\mathrm{v}} < 3$ (Matsuda et al., 2014).

For the cases of polydisperse particles, the settling influence on the coherence term must be considered as well as the influence on $E^*_{\mathrm{np}}(\xi|\mathrm{St})$ in Eq. (33). Ayala et al. (2008a) and Lu et al. (2010) reported that gravitational settling enlarges the cross-over length of the bidisperse RDF. Lu et al. (2010) extended the perturbation expansion analysis of Chun et al. (2005) and presented the formulation for the cross-over length in the presence of gravity, which is

$$\frac{l_{\mathrm{c}}}{l_{\eta}} = C_{\mathrm{Chun}} |\mathrm{St}_1 - \mathrm{St}_2| \left[ 1 + \frac{1}{3a_0} \left( \frac{\tau_{\mathrm{g}}}{\tau_{\mathrm{a}}} \right) \mathrm{Fr}^{-2} \right]^{1/2}, \tag{38}$$

where $C_{\mathrm{Chun}}$ is the coefficient derived by Chun et al. (2005), $a_0$ is the ratio of the acceleration variance $\langle a^2 \rangle$ to square of the Kolmogorov acceleration $a_{\eta}^2$, $\tau_{\mathrm{a}}$ is the acceleration correlation time scale, and $\tau_{\mathrm{g}}$ is the correlation time scale for gravitational settling particles. Chun et al. (2005) obtained the values of $C_{\mathrm{Chun}} \approx 5.0$ and $a_0 \approx 1.545$ based on their DNS results. Lu et al. (2010) further assumed $S_{\mathrm{v}} \lesssim 1$ so that $\tau_{\mathrm{g}}$ is simply given by $\tau_{\mathrm{g}} = \tau_{\mathrm{
[revised manuscript text omitted]
_{\mathrm{n}}(k)}{\chi_{\mathrm{n}} \epsilon^{-3/4} \nu^{5/4}} = C_{\mathrm{c}}(kl_\eta)^{-5/3} \exp\left[-1.5 \frac{C_{\mathrm{c}}}{\mathrm{Pr}} (kl_\eta)^{4/3}\right], \qquad (48)$$

where $\chi_{\mathrm{n}}$ is the scalar dissipation rate for $n_{\mathrm{ref}}$. In this study, the Prandtl number of refractive index was set to 0.7.

The scalar dissipation rates for $\langle r_{\mathrm{p}}^3 \rangle n_{\mathrm{p}}$ and $n_{\mathrm{ref}}$ were calculated in the same way. That is, the dissipation rate of an arbitrary scalar $\theta$ is given by

$$\chi_\theta = 2 \frac{\nu_{\mathrm{t}}}{\mathrm{Sc}_{\mathrm{t}}} \frac{\partial \widehat{\theta}}{\partial x_i} \frac{\partial \widehat{\theta}}{\partial x_i}, \qquad (49)$$

10 where $\nu_{\mathrm{t}}$ is the eddy kinematic viscosity, $\mathrm{Sc}_{\mathrm{t}}$ is the turbulent Schmidt number, and $\widehat{\theta}$ is the scalar value in the resolved scale. $\nu_{\mathrm{t}}$ was calculated by using the Smagorinsky model, i.e., $\nu_{\mathrm{t}} = (C_{\mathrm{s}}\Delta_{\mathrm{s}})^2 (2\widehat{s_{ij}}\widehat{s_{ij}})^{1/2}$, and $\mathrm{Sc}_{\mathrm{t}}$ was set to 0.4 (Moin et al., 1991).

**5.3 Results and discussion**

Figure 8 (a) shows the three-dimensional visualization of liquid water. The optical thickness of each grid cell, $\tau_\Delta$, is vi-
15 sualized by volume rendering to mimic human-eye observations of clouds. Here, the optical thickness is defined by $\tau_\Delta = Q_{\mathrm{ext}}\pi\langle r_{\mathrm{p}}^2\rangle n_{\mathrm{p}}\Delta_{\mathrm{z}}$, where $Q_{\mathrm{ext}}$ is the extinction efficiency for Mie scattering ($Q_{\mathrm{ext}} = 2.0$ in this study), and $\langle r_{\mathrm{p}}^2\rangle$ is gven by $\langle r_{\mathrm{p}}^2\rangle = \int_0^\infty r_{\mathrm{p}}^2 q_{\mathrm{r}}(r_{\mathrm{p}})dr_{\mathrm{p}}$. Note that the optical transmittance of cloud volume is approximately equal to $1 - \tau_\Delta$ when $\tau_\Delta$ is sufficiently smaller than unity. Figure 8 (b) shows the isosurfaces of the energy dissipation rate $\epsilon$ and the vertical velocity $u_3$. The locations of upward flows correspond to the locations of clouds and large $\epsilon$ regions are observed around the upward flows.
20 This indicates that strong turbulence is generated by entrainment motions due to updrafts.

This study focused on a vertical cross section that slices the cumulus cloud with the largest upward velocity. Figure 9 shows the liquid water content (LWC) in a logarithmic scale, the energy dissipation rate $\epsilon$, the radar reflectivity factor $Z^{\mathrm{dB}}$, the increase of $Z^{\mathrm{dB}}$ due to particulate Bragg scattering, and particulate Bragg scattering due to turbulent clustering $Z^{\mathrm{dB}}_{\mathrm{PBc}}$ in the cross section. The microwave frequency was set to $f_{\mathrm{m}} = 2.8\,\mathrm{GHz}$, which is the representative frequency of S-band radars. The
25 radar reflectivity factor is shown in units of decibels, which is defined as $Z^{\mathrm{dB}}\,(\mathrm{dBZ}) = 10\log_{10} Z(\mathrm{mm}^6/\mathrm{m}^3)$. Large values of $Z^{\mathrm{dB}}$ are observed inside and below the clouds. The strong echo below the clouds reflects drizzling regions, where the LWC is smaller than that inside the clouds but the strong echo returns from the drizzling droplets because $Z_{\mathrm{incoh}}$ is proportional to $\langle r_{\mathrm{p}}^6\rangle$. The radar echo at outside of the isoline of $Z^{\mathrm{dB}}_{\mathrm{incoh}} = -18\,\mathrm{dBZ}$ is caused by clear-air Bragg scattering; i.e., $Z_{\mathrm{CB}}$. The radar echo layer at the height from 2.2 to 2.5 km is caused by $Z_{\mathrm{CB}}$ due to a large humidity gap in the inversion layer.
30 Figure 9 (c) does not show a clear sign of the mantle echo reported by Knight and Miller (1998). Knight and Miller (1998) reported that the predominant mantle echo is observed on dry days, while it is poorly observed on the most humid day. Our result is in accord with this report as the relative humidity of the environmental air is above 80% at heights below about 2.2 km

[Figure]

(a)

(b)

**Figure 8.** Three-dimensional visualization of cloud simulation data: (a) volume rendering of optical depth and (b) isosurfaces of (blue) the energy dissipation rate $\epsilon = 100$ cm$^2$/s$^3$ and (yellow) the vertical velocity $u_3 = 3$ m/s.

in our cloud simulation data. A possible cause of the mantle echo is particulate Bragg scattering due to turbulent entrainment (i.e., $Z_{\mathrm{PBe}}$) because the large-scale cloud water inhomogeneity at cloud edges produces small-scale fluctuations due to the turbulent cascade (Erkelens et al., 2001). The influence of turbulent entrainment on $Z^{\mathrm{dB}}$ turned out to be, however, negligibly small. That is, the fluctuations caused by the large-scale inhomogeneity were not significantly large at the scale of the half

5    wavelength in the present simulation.

[Figure]

**Figure 9.** Liquid water content (LWC), energy dissipation rate $\epsilon$, and radar reflectivity factors for S-band microwaves in the vertical cross section; (a) LWC in a logarithmic scale (i.e., $\log_{10} \text{LWC} (\text{g/m}^3)$), (b) $\epsilon$ $(\text{m}^2/\text{s}^3)$, (c) $Z^{\text{dB}}$ (dBZ), (d) $
[revised manuscript text omitted]